# Structure of the Signal Transduction Domain in Second-Generation CAR Regulates the Input Efficiency of CAR Signals

**DOI:** 10.3390/ijms22052476

**Published:** 2021-03-01

**Authors:** Kento Fujiwara, Masaki Kitaura, Ayaka Tsunei, Hotaka Kusabuka, Erika Ogaki, Naoki Okada

**Affiliations:** Project for Vaccine and Immune Regulation, Graduate School of Pharmaceutical Sciences, Osaka University, 1-6 Yamadaoka, Suita, Osaka 565-0871, Japan; fujiwara-k@phs.osaka-u.ac.jp (K.F.); kitaura-m@phs.osaka-u.ac.jp (M.K.); ktkn0810@gmail.com (A.T.); hkusabuka@gmail.com (H.K.); oogaki1116@gmail.com (E.O.)

**Keywords:** chimeric antigen receptor, signal transduction domain, CAR structure-activity relationship

## Abstract

T cells that are genetically engineered to express chimeric antigen receptor (CAR) have a strong potential to eliminate tumor cells, yet the CAR-T cells may also induce severe side effects due to an excessive immune response. Although optimization of the CAR structure is expected to improve the efficacy and toxicity of CAR-T cells, the relationship between CAR structure and CAR-T cell functions remains unclear. Here, we constructed second-generation CARs incorporating a signal transduction domain (STD) derived from CD3ζ and a 2nd STD derived from CD28, CD278, CD27, CD134, or CD137, and investigated the impact of the STD structure and signaling on CAR-T cell functions. Cytokine secretion of CAR-T cells was enhanced by 2nd STD signaling. T cells expressing CAR with CD278-STD or CD137-STD proliferated in an antigen-independent manner by their STD tonic signaling. CAR-T cells incorporating CD28-STD or CD278-STD between TMD and CD3ζ-STD showed higher cytotoxicity than first-generation CAR or second-generation CARs with other 2nd STDs. The potent cytotoxicity of these CAR-T cells was not affected by inhibiting the 2nd STD signals, but was eliminated by placing the STDs after the CD3ζ-STD. Our data highlighted that CAR activity was affected by STD structure as well as by 2nd STD signaling.

## 1. Introduction

Chimeric antigen receptor (CAR) is a fusion protein of a tumor antigen-binding molecule and a T cell-activating molecule. Transduction of CAR gene into T cells is a convenient method for creating the tumor cell-specific cytotoxic T cells (CAR-T cells). Adoptive immunotherapy using CAR-T cells have become an innovative treatment option for patients with refractory or recurrent B-cell malignancies [1,2,3,4]. Unfortunately, life-threatening side effects, such as cytokine release syndrome and injury to normal organs essential for life support, have been observed in clinical trials [5,6,7]. These side effects, not only threaten patient safety, but also limit the effectiveness of CAR-T cell therapy. Since the occurrence of adverse events is thought to be caused by the high reactivity and excessive activation of CAR-T cells [8,9,10], improvements in the quality and performance of CAR-T cells are urgently required for this therapy to fulfill its promise as a safe treatment option for refractory cancer patients.

The functionality of CAR-T cells depends on the CAR expression level and its activity on the cell membrane [11,12,13]. CAR consists of four regions from the extracellular domain of the antigen receptor: the antigen recognition domain (ARD), the hinge domain (HD), the transmembrane domain (TMD), and the intracellular signal transduction domain (STD). CAR-T cell activation is induced by binding of the target antigen via the ARD and subsequent dynamic conformational change of the STD. Since the first report in 1993 of CAR prototypes that consisted of a tandemly linked single-chain variable fragment (scFv) and a CD3ζ moiety [14], CAR-T cell performance has been steadily enhanced. This has been achieved by modification of CAR components, such as improving CAR expression stability on the cell membrane by changes to HD/TMD, and strengthening of CAR-T cell proliferation and anti-tumor activity by the addition of a co-stimulatory molecule-derived STD [13]. Further refinements in the design of CAR structures are expected to lead to reduced toxicity and enhanced anti-tumor activity in CAR-T cells [15]. Although a variety of CAR structures have been developed, these CARs are unique to the researchers, and there is a lack of studies that have analyzed the linkage between the structure and activity of artificial CAR proteins in detail [15,16]. There has only been limited knowledge has been accumulated about the structural design of CARs to fine-tune CAR-T cell functions.

To mimic the complete activation mechanism of T cells, which requires co-input of a T-cell receptor signal (CD3 signal) and co-stimulatory molecular signals, structural studies on the STD (among other CAR components) have been eagerly pursued. Compared to the first-generation CAR-T cells with only CD3ζ-STD (1st STD), the second-generation CAR-T cells with CD3ζ-STD and 2nd STD derived from various co-stimulatory molecules are superior in their abilities to proliferate, produce cytokines and to persist in vivo, and they exhibit more potent anti-tumor activity [17,18,19,20,21,22]. The second-generation CARs have been developed by incorporating intracellular regions of proteins in the CD28 family, such as CD28 and CD278 (ICOS) or in the tumor necrosis factor receptor superfamily (TNFRSF) such as CD137 (4-1BB), CD134 (OX40), and CD27 as the 2nd STD. CARs with the 2nd STD derived from members of the CD28 family yield CAR-T cells that have enhanced proliferative activity and interleukin (IL)-2 secretion by signaling through phosphoinositide 3-kinase (PI3K) [23,24]. CARs with the 2nd STD derived from members of the TNFRSF enhance in vivo persistence of CAR-T cells by signaling through TNFR-associated factors (TRAFs) [24,25]. However, the characteristics of these second-generation CAR-T cells have been evaluated by comparing them with first-generation CAR-T cells or second-generation CAR-T cells that differ in HD/TMD as well as STD [26,27]. The detailed impact of the addition of the 2nd STD on CAR activity and CAR-T cell function has not been clarified. Given that the activation strength of CAR-T cells via CAR depends on the CAR surface expression level and its signal input efficiency based on the HD/TMD structure [11], the difference in functionality of those second-generation CAR-T cells, shown in previous studies, may be a complex result reflecting not only the effect of the added 2nd STD signal, but also the effect of the CAR expression level and the signal input efficiency of CD3ζ-STD. Therefore, it is desirable to fully characterize the factors in the 2nd STD that affect CAR activity to aid in fine-tuning CAR-T cell performance by modifying STD structure.

We have previously shown that the vascular endothelial growth factor receptor 2 (VEGFR2)-specific first-generation CAR (V/28/28/3z) with CD28-HD/TMD is stably expressed as a homogeneous molecule on mouse T cell membranes [11]. In this study, we constructed STD-modified second-generation CARs based on CAR (V/28/28/3z) by incorporating a 2nd STD derived from CD28 family molecules or TNFRSF molecules between TMD and CD3ζ-STD or at the end of CD3ζ-STD. We analyzed the CAR expression and activity in mouse CAR-T cells. We also evaluated the effects of the different types of STD derived from co-stimulatory molecules and the order of their insertion on the expression and activity of the CARs.

## 2. Results

### 2.1. Expression of Various Second-Generation CARs on Mouse T Cells

To profile the expression and functional characteristics of second-generation CARs incorporating co-stimulatory molecules with different signaling properties in T cells, we constructed five types of second-generation CARs by inserting STDs derived from CD28 family members (CD28 or CD278) or from TNFRSF members (CD27, CD134, or CD137) between CD28-TMD and CD3ζ-STD in a first-generation CAR (V/28/28/3z) reactive to mouse VEGFR2 (Figure 1A). Mouse T cells were transfected with retroviral vectors (Rv) carrying these CAR genes, and the expression of second-generation CAR mRNA in the T cells was comparable to that of first-generation CARs. The expression level of the second-generation CAR mRNAs remained constant for at least 6 days after Rv transduction (Figure 1B). Four second-generation CARs, except V/28/28/137-3z, were efficiently expressed on T cell membranes after Rv transduction, though their expression levels were slightly lower than that of V/28/28/3z (Figure 1B, Appendix AA). The disappearance profile of these second-generation CARs from the cell surface was similar to that of V/28/28/3z, and these second-generation CARs were highly expressed on T cells even 6 days after Rv transduction. However, only V/28/28/137-3z showed a low expression level on the cell surface after Rv transduction, and its expression was hardly detected 2 days later. These results indicated that depending on the structure of the 2nd STDs to be added, the CAR expression efficiency and stability on the membrane might be lowered.

Most of the CARs with CD137-STD, which are frequently used in human CAR-T cell research, utilize CD8α-HD/TMD instead of CD28-HD/TMD [18,20,26,27]. Therefore, we hypothesized that the combination of CD28-TMD and CD137-STD might affect the interaction between the intracellular region of CAR and the plasma membrane lipids, resulting in poor expression of V/28/28/137-3z. We, thus, constructed CAR (V/8a/8a/137-3z) that was modified from CD28-HD/TMD to CD8α-HD/TMD, and CARs (V/28/28/Δ137-3z and V/8a/8a/Δ137-3z) that incorporated ΔCD137-STD by deleting 20 amino acids near the TMD, which contains many cationic amino acids, while preserving the signal input motif of CD137-STD (Figure 1C, Appendix AB). The expression of these CARs with CD137-STD in mouse T cells was analyzed. All CAR constructs showed CAR mRNA levels comparable to those of V/28/28/3z for at least 6 days after Rv transduction, confirming that HD/TMD and STD modifications do not affect the transcriptional process of CAR genes (Figure 1D). The surface expression levels of these CARs were improved by modification of CD8α-HD/TMD and deletion of amino acids near the TMD in the CD137-STD sequence (Figure 1D). Among the CARs examined, V/28/28/Δ137-3z showed the highest CAR expression level. The expression level of V/28/28/Δ137-3z was inferior to the membrane expression intensity of the first generation CARs, but improved to CAR expression levels comparable to those of other second-generation CARs. Interestingly, V/8a/8a/Δ137-3z, which was modified with both HD/TMD and CD137-STD, showed a lower expression level than V/28/28/Δ137-3z throughout the period from Rv transduction. These results highlighted that the combination of TMD and STD, as well as the respective sequences of TMD and STD affect the membrane expression efficiency of CAR proteins.

Taken together, these results suggested that the additional insertion of the 2nd STD after the TMD affects the surface expression level of CAR proteins. Moreover, the combination of TMD and STD and the exact sequence of STD near the TMD had a significant effect on the membrane expression efficiency of CAR. In the subsequent experiments, we used V/28/28/Δ137-3z as a second-generation CAR with CD137-STD, whose expression level in T cells was comparable to other second-generation CARs.

### 2.2. Function of Mouse T Cells Expressing Various Second-Generation CARs

We evaluated the proliferative activity, cytokine secretion, and cytotoxic activity of various second-generation CAR-T cells cultured for 4 days after Rv transduction (Figure 2, Appendix A). Analysis of the proliferative activity of each CAR-T cell without stimulation showed that two types of second-generation CAR-T cells (V/28/28/278-3z and V/28/28/Δ137-3z) showed higher proliferative activity than mock-transduced T cells and first-generation CAR-T cells (Figure 2A). Thus, these CAR-T cells were found to have CAR-mediated tonic signaling. The antigen dependent proliferative activity of all second-generation CAR-T cells increased in a VEGFR2-Fc density-dependent manner, similar to that of first-generation CAR-T cells (Figure 2A).

Cytokine secretion of each CAR-T cell was first observed upon VEGFR2-Fc stimulation, and all the second-generation CAR-T cells showed a greater ability to secrete interferon (IFN)-γ than the first-generation CAR-T cells (Figure 2B). Furthermore, V/28/28/278-3z and V/28/28/Δ137-3z showed an enhanced ability to secrete TNF-α and IL-2, and V/28/28/28-3z showed an enhanced ability to secrete IL-2. These results suggested that signaling via CD28-STD, CD278-STD, and ΔCD137-STD modulates the cytokine secretion ability of T cells. V/28/28/27-3z and V/28/28/134-3z were expected to augment the secretion of TNF-α in T cells by activating the 2nd STD-mediated nuclear factor-κB pathway, but they showed no superiority over first-generation CAR. Therefore, it was considered that V/28/28/27-3z and V/28/28/134-3z may not provide the appropriate 2nd STD signals.

All second-generation CAR-T cells specifically killed VEGFR2-expressing tumor cells, but their cytotoxic activity according to the effector cell to target cell (E/T) ratio differed greatly depending on the type of co-stimulatory molecule inserted as the 2nd STD (Figure 2C). V/28/28/28-3z and V/28/28/278-3z with the 2nd STD derived from members of the CD28 family had significantly higher cytotoxic activity per CAR expression level than V/28/28/3z, and there was no other difference between these two second-generation CAR-T cells. The second-generation CAR-T cells with STDs derived from members of the TNFRSF showed weaker cytotoxic activity than the first-generation CAR-T cells under conditions of a low E/T ratio, but showed stronger cytotoxic activity than the first generation CAR-T cells with an E/T ratio of 20. These results suggested that the CAR ability to induce CD3ζ-STD signaling-mediated cytotoxic activity was altered by differences in the type of 2nd STD (structure and signal input mechanism).

Collectively, these results indicated that insertion of CD28-STD and CD278-STD after the TMD enhances IFN-γ and IL-2 secretion and antigen-specific cytotoxic activity. The insertion of CD27-STD, CD134-STD, and ΔCD137-STD after the TMD enhanced IFN-γ secretion and raises the CAR activation threshold. Furthermore, depending on the type of additional STD inserted, CAR-mediated tonic signaling was found to be increased.

### 2.3. Expression and Function in Mouse T Cells of CAR Mutants Deficient in the Co-Stimulatory Signal Input Motif

Generally, the functional enhancement of second-generation CAR-T cells is considered to be due to the signaling from co-stimulatory molecules incorporated as the 2nd STD of CAR [13,22]. To evaluate the effect of the 2nd STD signals on CAR activity, we constructed second-generation CAR genes with a mutated 2nd STD in which the binding motif of the adapter molecule in each co-stimulatory molecule was replaced by other amino acids (Figure 3A). These CAR genes were transduced into mouse T cells by the Rv transduction method, and all the second-generation CARs deficient in the 2nd STD signal were expressed on the T cell membrane. The surface expression levels of these mutant CARs were comparable to those of the unmodified second-generation CARs (Figure 3B), and the membrane expression profiles of these CARs were not markedly different (data not shown).

We evaluated the proliferative activity of second-generation CAR-T cells lacking the 2nd STD signal, and found that none of the T cells expressing deficient in the 2nd STD signal input motif showed antigen-independent proliferation (Figure 3C). Accordingly, we attributed the antigen-independent proliferation of V/28/28/278-3z and V/28/28/137-3z expressing T cells to the 2nd STD signal. Although all CAR-T cells deficient in the 2nd STD signal responded to VEGFR2-Fc stimulation, the high proliferative activity of the unmodified second-generation CARs with CD28-STD, CD278-STD, or ΔCD137-STD was canceled by the lack of their signal input (Figure 3C). These results indicated that the input of the CD28, CD278, and CD137 signals enhanced the proliferative activity of CAR-T cells upon antigen stimulation. As in T cells expressing unmodified second-generation CARs, T cells expressing CARs deficient in the 2nd STD signal input motif secreted several cytokines upon antigen stimulation for the first time (Figure 3D, Appendix AA). Furthermore, we found that the enhancement of IFN-γ secretion by addition of the 2nd STD was unexpectedly observed in CARs deficient in the 2nd STD signal input motif as well (Figure 3D). In contrast, a lack of signal input motifs in CD28-STD, CD278-STD, or CD137-STD eliminated the enhancement of IL-2 and/or TNF-α secretion (Figure 3D). The ability of second-generation CARs, with CD27-STD or CD134-STD, to induce cell proliferation activity and cytokine secretion was not affected by the lack of their signaling motifs. These results suggested that the enhancement of TNF-α and IL-2-secreting ability of CAR-T cells depends on the signal input from the CD28-STD, CD278-STD, or CD137-STD, while the enhancement of IFN-γ secretion ability is affected by the structure of the CAR intracellular region due to insertion of the 2nd STD.

With the exception of CAR [V/28/28/27^Mut^-3z]-T cells, the four types of second-generation CAR-T cells lacking the 2nd STD signal exhibited a cytotoxicity profile comparable to that of unmodified second-generation CAR-T cells (Figure 3E, Appendix AB). Namely, the second-generation CARs with inserted the 2nd STD derived from members of the CD28 family showed high cytotoxic activity regardless of the presence or absence of signaling by the 2nd STD, while the second-generation CARs with CD134-STD or ΔCD137-STD inserted showed a high threshold of cytotoxicity but showed strong cytotoxicity activity when the E/T ratio was high. In an unexpected result, V/28/28/27^Mut^-3z showed a higher ability to induce cytotoxic activity than V/28/28/27-3z at all E/T ratios, and its level was comparable to that of second-generation CARs with CD28-STD or CD278-STD.

Collectively, these results indicated that the enhanced secretion of TNF-α and IL-2, and antigen-independent cell proliferation by CAR-mediated tonic signaling in second-generation CAR-T cells are brought about by the input of 2nd STD signals. In contrast, the enhancement in IFN-γ secretion and cytotoxic activity of the second-generation CAR-T cells was a small contribution of the 2nd STD signal.

### 2.4. Expression and Function in Mouse T Cells of Second-Generation CARs with an Altered Order of STDs

The results showed that the cytotoxic activity of the second-generation CAR-T cells changed depending on the type of 2nd STD, while it was not affected by the 2nd STD signal, suggesting that the signal input efficiency of the CD3ζ-STD located at the C-terminal end of the CAR changed depending on the structure (2nd STD structure) between the TMD and CD3ζ-STD. We, thus, constructed CARs whereby the STDs were reordered such that each 2nd STD was not inserted after the CD28-TMD of V/28/28/3z but was linked to the end of the CD3ζ-STD (Figure 4A). We then analyzed the effect of STD structure on CAR expression and activity. The expression level in mouse T cells of the CARs with reordered STDs was comparable to that of the unmodified second-generation CARs (Figure 4B), and the expression profiles of these CARs were also not different (data not shown). This result indicated that reordering of the STDs did not affect the CAR expression efficiency.

Among the CAR-T cells with reordered STDs, only the CAR-T cells with ΔCD137-STD showed antigen-independent proliferative activity even when the STD order was changed (Figure 4C). In contrast, CD278-STD-mediated tonic signaling was observed only in V/28/28/278-3z, suggesting that CD278-STD in V/28/28/278-3z cannot maintain the same conformation as endogenous CD278. This finding highlighted the importance of the combination of TMD and 2nd STD, and the design of STD structure. The proliferative activity to VEGFR2-Fc stimulation in all CAR-T cells with reordered STDs was comparable to that of the unmodified second-generation CAR-T cells.

All the CAR-T cells with reordered STDs secreted larger amounts of IFN-γ than the first-generation CAR-T cells, though the IFN-γ secretion ability of CARs with CD28-STD and CD278-STD was decreased by modifying the order of the STDs (Figure 4D, Appendix AA). In addition, regardless of the order of the STDs, CARs with STDs derived from members of the CD28 family showed higher IL-2 secretion and CARs with ΔCD137-STD showed higher TNF-α secretion. Thus, the CARs with STDs derived from CD28, CD278, and CD137 can be expressed on the plasma membrane as a structure that allows the 2nd STD signal to be input even if the insertion position of the 2nd STD is changed. The results also suggested that the signal input efficiency of CARs with CD28-STD and CD278-STD may be affected by the order of the 2nd STD and CD3ζ-STD.

Interestingly, the CAR-T cells with CD28-STD or CD278-STD after the CD3ζ-STD had significantly reduced cytotoxic activity when compared to the unmodified second-generation CAR-T cells (Figure 4E, Appendix AB). In the second-generation CAR-T cells with STDs derived from members of the TNFRSF, the change in STD order had no effect on cytotoxic activity. Therefore, the change in the entire STD structure associated with the insertion of CD28- and CD278-STD between TMD and CD3ζ-STD was found to be the main factor that enhanced the cytotoxic activity of V/28/28/28-3z and V/28/28/278-3z.

Taken together, these results indicated that the enhanced secretion of IFN-γ in CAR-T cells was based on the lengthening of the whole STD associated with the insertion of the 2nd STD, while the CAR-mediated tonic signaling and the enhanced secretion of TNF-α and IL-2 were based on the signaling input of the 2nd STD. Furthermore, the position of the 2nd STD in second-generation CARs had a significant effect on the cytotoxic activity of CAR-T cells, suggesting that addition of the 2nd STD after TMD may alter the signal input efficiency of the CD3ζ-STD depending on its structure.

## 3. Discussion

In this study, we conducted a systematic analysis of the relationship between structure and activity of various second-generation CARs with shared ARD/HD/TMD, and found that the differences in STD structures in second-generation CARs affect the expression efficiency of CARs and the functionality of CAR-T cells. The functional enhancement of the second-generation CAR-T cells as compared to the first-generation CAR-T cells was largely attributed to the added second STD signal [13,22]. As summarized in Figure 5A, the co-stimulatory signals derived from the 2nd STD enhanced proliferation of CAR-T cells and increased secretion of TNF-α and IL-2, consistent with previous reports [20,22,24,26]. In addition, we found that the cytotoxic activity of second-generation CAR-T cells was affected by the structure of the STD rather than the signal input from the 2nd STD. This finding suggested that the selection of an appropriate co-stimulatory signal as well as the structural design of the entire CAR intracellular domain, including the structure and additional position of the 2nd STD, should be considered in efforts to modify the characteristics of CAR-T cells. To further optimize the performance of CAR-T cells by modification of the STD, it is desirable to extract the structural elements of STD that affect the expression and activity of CAR and to further clarify the relationship between CAR structure and activity by focusing on these elements.

Since the responsiveness of CAR-T cells to target antigen density is thought to be defined by the affinity and avidity of the CAR, regulation of CAR expression is a key approach to modulating the performance of CAR-T cells [11,12,13]. The expression efficiency of second-generation CARs, except for V/28/28/137-3z, on T cell membranes was slightly lower than that of first-generation CARs, but was not affected by mutations of signal input motifs or changes in order of the STDs. This suggests that the factor responsible for the reduced membrane expression level of the second-generation CAR is the lengthening of the whole STD structure by the addition of the 2nd STD. Furthermore, a series of expression analyses of CARs with CD137-STD revealed that the combination of TMD and STD, and presumably the sequence around the TMD-STD linkage, affects the expression efficiency of CAR and its stability on the membrane. Hydrophilic amino acids near the α-helix domain as well as hydrophobic amino acids that form the α-helix domain in the TMD play an important role in stabilizing transmembrane proteins in the cell membrane [28]. Compared to the human CD137-STD, the mouse CD137-STD used in this study contains three extra amino acids (Ser-Val-Leu) at the N-terminal end adjoining the TMD (Appendix A). Given that there are no reports that human CARs with CD137-STD are refractory to expression [15,25,26,27], the presence of these non-polar amino acids may be a factor that reduces the expression efficiency of V/28/28/137-3z. Consequently, it should be confirmed beforehand that planned structural modifications of the STD do not disrupt the α-helix domain in the TMD and the conserved structure of the STD. A membrane protein structure prediction system such as TMHMM [29] would be a useful tool for this purpose.

Tonic signaling of CARs expressed on T-cell membranes has been shown to affect the quality and in vivo persistence of CAR-T cells, warranting an investigation into the cause and to offer possible solutions [25,30,31]. In the present study, we evaluated the proliferative activity of CAR-T cells and found that the addition of some types of 2nd STDs resulted in CARs with increased ligand-independent tonic signaling. Previous studies using human second-generation CAR with CD137-STD reported that constant input of CD137-STD-mediated TRAF2 signaling led to antigen-independent proliferation of CAR-T cells [25]. Although our analysis was based on a second-generation CAR with a CD137-STD lacking 20 amino acids near the TMD, we showed that the antigen-independent proliferation of CAR-T cells was due to ΔCD137-STD-mediated tonic signaling, and furthermore that ΔCD137-STD-mediated tonic signaling could not be canceled even if the order of the STDs was swapped. This may suggest that the second-generation CARs with ΔCD137-STD have an STD structure that facilitates the formation of TRAF2-mediated signalosomes regardless of the insertion position of ΔCD137-STD. To avoid CAR tonic signaling, it will be necessary to analyze the structure of the CAR intracellular region and develop an understanding of its conformation and interaction with the cell membrane. The redesign of the CAR intracellular region based on structural knowledge should allow the input of STD signals only when the conformational change of the CAR intracellular region occurs upon ligand binding.

CD3ζ-STD signaling in first-generation CAR is assumed to be input into the cell by a mechanism similar to that of the T-cell receptor and CD3 complex [32]. Mechanical stimulation through binding of the ARD to the antigen releases into the cytoplasm the immunoreceptor tyrosine-based activation motif (ITAMs) in CD3ζ-STD, which are protected by the membrane at rest. Subsequent phosphorylation of ITAMs by lymphocyte-specific protein tyrosine kinase (Lck) and binding of Zap70 to phosphorylated ITAMs triggers T cell activation [33]. Although the same CD3ζ-STD signaling mechanism was expected to induce CAR activation in second-generation CARs, the present study revealed that the efficiency of CD3ζ-STD signaling depends on the position and structure of the 2nd STD. In earlier CAR studies, it has been reported that the order of CD3ζ-STD and CD28-STD affects the effector function of CAR-T cells [34,35]. Although it is not possible to determine from these studies whether the functional strength of CAR-T cells was affected by the efficiency of CAR expression or STD structure, these studies are important in supporting our findings. Furthemore, a recent study using phosphoproteome analysis of human second-generation CAR-T cells with CD28-STD or CD137-STD revealed that CAR with CD28-STD input strong activation signals in a shorter time than CAR with CD137-STD [24]. Our results are consistent with this report, showing that the insertion of a CD28-STD and CD278-STD after the TMD improves the efficiency of CD3ζ-STD signal input. Although the phosphorylation efficiency of CD3ζ-STD in CAR with CD28-STD has been thought to be possibly related to the binding of Lck in CD28-STD [24], our analysis using CARs with CD28-STD lacking either PI3K-binding or Lck-binding motifs suggested that the Lck-binding in CD28-STD does not significantly affect the phosphorylation efficiency of CD3ζ-STD (Appendix A). We hypothesize that the strong signal input intensity in CARs with CD28-STD and CD278-STD is due to the ease with which CD3ζ-STD is exposed to the cytoplasm. Previous studies analyzing the presence of CD28 on the cell membrane have shown that the CD28 tail, like CD3ζ, protects phosphorylation in signaling motifs by lining the cell membrane [36]. Thus, the interaction of the 2nd STD with the plasma membrane may have affected the input efficiency of CD3ζ-STD signaling (Figure 5B).

Unlike CD28 and CD278, TNFRSF inputs TRAF signaling by trimerization [37]. This difference in the signal input mechanism implies that the intracellular structure and their presence state are distinct, which may have affected the input efficiency of CAR signal (Figure 5B). Recent study has shown that the activation intensity of CAR-T cells varies according to the activation sites of the three ITAMs in CD3ζ-STD [38]. The structural change of the entire STD due to the addition of the 2nd STD may have affected the phosphorylation efficiency of CD3ζ-STD. The second-generation CAR with CD137-STD has been suggested to negatively regulate the phosphorylation of CD3ζ-STD by recruiting the THEMIS-SHP1 complex upon signal input [39]. In this study, we found that second-generation CAR with CD27-STD did not enhance TNF-α secretion by CD27-STD signaling, while the loss of TRAF-binding motifs induced high cytotoxic activity, although the cause of this effect was not clarified in detail. These findings suggest that the TNFRSF-derived STD in the second-generation CAR shows a different structural state from that of endogenous TNFRSF and may interact with molecules other than TRAFs. Based on the above results and reports, we hypothesized that the efficiency of CD3ζ-STD signal input in CARs is affected by the insertion of the 2nd STD due to differences in its structure and signal characteristics (Appendix A). We believe that further analysis of CARs with artificial 2nd STD without signal input motifs, as well as biochemical analysis of signaling pathways and structural analysis of intracellular regions in second-generation CARs, will help us to elucidate the elements of STD that affect the signal input efficiency of CARs as shown in Figure 5B.

In this study, we standardized several functional intensities of CAR-T cells at the level of CAR expression to closely examine the effect of STD structure on CAR expression and activity. As a result, we found that the signal input efficiency of CARs varied depending on the type and position of the 2nd STD. The expression level of CARs is expected to vary depending not only on the CAR structure but also on the method of CAR gene introduction [40]. The recent development of gene transfer technology and genome editing technology utilizing DNA plasmids and viral vectors has made it possible to create efficient cellular drugs and also control the amount and location of gene incorporation into cells. Based on the findings of our study, we expect to create more functional CAR-T cells by selecting second-generation CARs with desired activity, appropriate gene delivery methods, and the number of incorporated copies.

In summary, this study revealed that the amino acid sequence near TMD in STD affected the efficiency of CAR expression and its stability on the cell surface. Moreover, while the co-stimulatory signals of the CD28-STD, CD278-STD, and CD137-STD enhanced the secretion of TNF-α and/or IL-2 in T cells depending on their type, the structure of the 2nd STD and the position of its incorporation into the CD3ζ-STD affected the IFN-γ secretion and cytotoxic activity of CAR-T cells. We highlight that the insertion of STD from members of the CD28 family between TMD and CD3ζ-STD lowers the CD3ζ-STD signal input threshold and strongly induces CAR-T cell activation. In the future, we hope to elucidate the structural factors of the entire STD that affect the signal input efficiency of CD3ζ-STD and to establish a method for fine-tuning the STD. This is a significant step towards creating highly functional CAR-T cells by designing a CAR structure that enables antigen-specific signal input of a 2nd STD derived from members of the TNFRSF.

## 4. Materials and Methods

### 4.1. Cell Lines

Human Plat-E cells were obtained from Cell Biolabs (San Diego, CA, USA) and cultured in Dulbecco’s modified Eagle’s medium supplemented with 10% fetal bovine serum (FBS, Thermo Fisher Scientific, Waltham, MA, USA), 1 μg/mL puromycin, and 10 μg/mL blasticidin. Murine EL4 cells were obtained from Cell Resource Center for Biomedical Research, Institute of Development, Aging, and Cancer, Tohoku University (Sendai, Japan), and cultured in RPMI 1640 medium supplemented with 10% FBS and 50 μM 2-mercaptoethanol. Murine VEGFR2-expressing EL4 (VEGFR2^+^ EL4) cells were generated via Rv transduction (containing the VEGFR2 gene and a puromycin resistance cassette) and were grown in the same culture medium as untransduced EL4 cells, supplemented with 5 μg/mL puromycin. All cells were maintained in a humidified atmosphere of 5% CO_2_ at 37 °C.

### 4.2. Mice

Female C57BL/6 mice (6–8-weeks old) were purchased from SLC (Hamamatsu, Japan) and were maintained in the experimental animal facility at Osaka University. Care and use of laboratory animals complied with the guidelines and policies of the Act on Welfare and Management of Animals in Japan. Protocols and procedures were approved by the Animal Care and Use Committee of Osaka University (approval number, douyaku 28-7-1; approval date, October 4th, 2017).

### 4.3. Construction of CAR Structural Variants

The HD/TMD and STD gene fragments of murine CD3ζ (BC052824.1), murine CD8α (BC030679.1), murine CD28 (BC064058.1), murine CD278 (BC034852.1), murine CD27 (BC171931.1), murine CD134 (BC139266.1), and murine CD137 (BC028507.1) were cloned from cDNAs derived from mouse splenocytes and lymph node cells. The STD gene fragments for the construction of the second-generation CARs were tandemly linked by the Gibson assembly with the STD genes linked in any order to each co-stimulatory molecule-derived 2nd STD and CD3ζ-STD. Furthermore, murine CD28-HD/TMD fragment and those STD fragments were assembled by Gibson assembly to include a SacII site at the 5′ end and a NotI site at the 3′ end. In second-generation CARs with CD137-STD, the mouse CD28- or CD8α-HD/TMD fragment and the STD fragment containing CD137 or ΔCD137 were assembled by Gibson assembly to include a SacII site at the 5′ end and a NotI site at the 3′ end.

We previously constructed pMXs-IG and pMXs-Puro (Cell Biolabs) carrying the anti-mouse VEGFR2 first-generation CAR gene [11]. The mouse VEGFR2-specific first-generation CAR construct contained an EcoRI restriction site, an Igκ-chain leader sequence, an HA-tag, an anti-mouse VEGFR2 scFv (clone avas12) [41], a SacII restriction site, murine CD28-HD/TMD, murine CD3ζ-STD, and a NotI restriction site. To construct the second-generation CARs, The HD/TMD/STD fragments were digested with SacII and NotI restriction enzymes (New England Biolabs, Ipswich, MA, USA) and individually ligated into similarly digested pMXs-IG containing the first-generation CAR (V/28/28/3z). The purified HD/TMD/STD gene fragment and pMXs-IG, from which HD/TMD/STD was removed, were joined using a DNA ligation kit (Takara Bio, Kusatsu, Japan) to construct the desired second-generation CAR gene-carrying pMXs-IG. The pMXs-IG carrying the 2nd STD signal input motif-deficient CAR gene was synthesized by inverse PCR using pMXs-IG carrying second-generation CARs as a template and a set of primers introducing mutations in the sequence of the signal input motif to produce linear pMXs-IG with a second-generation CAR gene. These genes were joined using a DNA ligation kit to construct pMXs-IG carrying the 2nd STD signal input motif-deficient CAR gene.

The VEGFR2-specific CARs that were generated are described as V/HD/TMD/STD. For example, CARs with CD28-STD added post-TMD are described as V/28/28/28-3z, and CD28-STD signal-deficient CAR are described as V/28/28/28^Mut^-3z. Sequence integrity of all plasmids was confirmed by DNA sequencing (Fasmac Co., Atsugi, Japan). The gene sequences and amino acid sequences of immune molecules used in this study are summarized in Table 1 and Table 2, respectively.

### 4.4. Production of CAR-T Cells

Murine CAR-T cells were produced as previously described [42]. Briefly, the Rv packaging CAR gene was produced by transfecting Plat-E cells with pMXs-Puro/CAR. Murine T cells were activated by using anti-CD3ε mAb (clone 145-2C11, Bioxcell, West Lebanon, NH, USA) and an anti-CD28 mAb (clone 37.51, Bioxcell), and then transduced with Rv-bound Retronectin (Takara Bio) under anti-CD3ε/CD28 mAbs stimulation. After Rv-transduction, CAR-T cells were cultured in RPMI 1640 medium supplemented with 10% FBS, 50 μM 2-mercaptoethanol, MEM Non-essential Amino Acids Solution (FUJIFILM Wako Pure Chemical, Osaka, Japan), 10 U/mL interleukin-2 (Peprotech, Rocky Hill, NJ, USA), and 5 μg/mL puromycin.

### 4.5. RT-qPCR Analysis for CAR mRNA Expression

Total RNA was isolated from CAR-T cells with TRIzol reagent (Thermo Fisher Scientific), and reverse-transcribed using Super Script III Reverse Transcriptase (Thermo Fisher Scientific). CAR cDNA was detected with the Custom TaqMan Gene Expression Assay (Thermo Fisher Scientific) for anti-VEGFR2 scFv. CAR and Gapdh expression levels were measured using the CFX96 Real-Time PCR Detection System (Bio-Rad Laboratories, Hercules, CA, USA). CAR mRNA level was normalized to the Gapdh mRNA level based on the comparative threshold cycle method (2^-ΔCt^).

### 4.6. Flow Cytometry Analysis for CAR Surface Expression

Murine CAR-T cells were incubated with an anti-mouse CD16/CD32 mAb (clone 93, Biolegend, San Diego, CA, USA), and then stained with the Zombie Aqua Fixable Viability Kit (Biolegend) and PE-Cy7-labeled anti-CD8α mAb (clone 53-6.7, Biolegend). CAR expression was evaluated using APC-labeled anti-HA.11 Epitope Tag mAb (clone 16B12, Biolegend) or APC-labeled mouse IgG1 isotype control mAb (clone MOPC-21, Biolegend). Immunofluorescence was measured using a BD FACS Canto II (BD Biosciences, Franklin Lakes, NJ, USA), and data were analyzed using FlowJo software (FlowJo LLC, Ashland, OR, USA). Data were represented as GMFI ratios, calculated according to the following formula: GMFI ratio = (GMFI of CAR-T cells stained with anti-HA mAb)/ (GMFI of CAR-T cells stained with mouse IgG1 isotype control mAb).

### 4.7. BrdU Proliferation Assay and Cytokine ELISA

CAR-T cells 4 days after Rv transduction were cultured for 24 h on a plate coated with VEGFR2-Fc (2–2000 ng/mL). Proliferation activity was measured by BrdU uptake ELISA (Sigma-Aldrich, St. Louis, Missouri, USA). The production of IFN-γ, TNF-α, and IL-2 in the supernatants was measured using the OptiEIA™ ELISA Set (BD Bioscience).

### 4.8. Cytotoxicity Assay

EL4 cells that do not express VEGFR2 were labeled with Tag-It Violet Proliferation Cell Tracking Dye (Biolegend), and VEGFR2^+^ EL4 cells were labeled with Cell Proliferation Dye eFluor 670 (Thermo Fisher Scientific). Mouse CAR-T cells 4 days after Rv transduction, EL4 cells, and VEGFR2^+^ EL4 cells were co-cultured at the indicated E/T ratios. After 18 h, CountBright Absolute Cell Counting Beads (Thermo Fisher Scientific) and 7-AAD Viability Staining Solution (Biolegend) were added to the reaction wells, and the number of each target cell was analyzed using flow cytometry until 1000 beads were detected. Cytotoxicity was calculated using the following formula: Percentage of antigen-specific lysis = [(VEGFR2^+^ EL4 cells/ EL4 cells ratio in non-effector cell’s well) − (VEGFR2^+^ EL4 cells/ EL4 cells ratio in effector cell’s well)] / [VEGFR2^+^ EL4 cells / VEGFR2^−^ EL4 cells ratio in non-effector cell’s well] × 100.

### 4.9. Statistical Analysis

All experimental data are represented as the mean ± *SD*. Statistical significance was evaluated using GraphPad Prism 8 Software (GraphPad Software, San Diego, CA, USA).

## Figures and Tables

**Figure 1 ijms-22-02476-f001:**
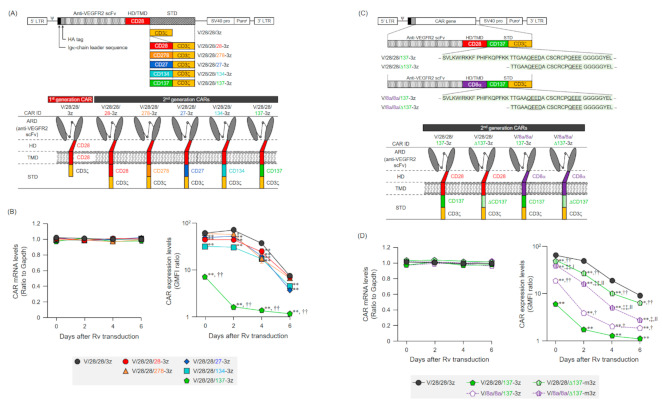
Expression of anti-mouse VEGFR2 second-generation CARs with different STDs. (**A**) Upper panel illustrates the Rv constructs containing the anti-mouse VEGFR2 first-generation CAR gene and second-generation CAR genes with a CD3ζ-STD (1st STD) and 2nd STD derived from different co-stimulatory molecules for mouse T cells. Lower panel represents STD-modified CAR proteins. (**B**) The left panel shows the results of reverse transcription and quantitative PCR (RT-qPCR) analysis of CAR mRNA. CAR mRNA expression level is represented relative to that of *Gapdh* mRNA. The data are shown as the mean ± SD of triplicates, and are representative of two independent experiments. Statistical analysis was performed using the Dunnett’s test for multiple comparisons with V/28/28/3z and showed no significant differences. The right panel shows the results of flow cytometric analysis of CAR expression on CD8^+^ T cells using anti-HA-tag mAb or isotype control antibodies. Expression level of each CAR was calculated from the ratio of GMFI when stained with the anti-HA-tag mAb to GMFI when stained with the isotype control antibody. The data shows the mean ± SD of three individual experiments. Statistical analysis was performed using the Dunnett’s test for multiple comparisons with V/28/28/3z: ** *p* < 0.01; and using the Tukey’s test with V/28/28/137-3z versus V/28/28/28-3z, V/28/28/278-3z, V/28/28/27-3z, and V/28/28/134-3z: ^††^
*p* < 0.01. (**C**) Upper panel illustrates the Rv constructs containing the anti-mouse VEGFR2 second-generation CAR genes with CD28- or CD8α-derived HD/TMD and CD137-derived STD or truncated CD137 (ΔCD137)-derived STD for mouse T cells. Lower panel shows representation of the corresponding CAR proteins. (**D**) The left panel shows the results of RT-qPCR analysis of CAR mRNA. CAR mRNA expression level is represented relative to that of *Gapdh* mRNA. The data are shown as the mean ± SD of triplicates, and are representative of two independent experiments. Statistical analysis was performed using the Tukey’s test for multiple comparisons and showed no significant differences. Right panel, CAR expression on T cells was analyzed by flow cytometry. The expression level of each CAR was calculated from the ratio of GMFI when stained with the anti-HA-tag mAb to GMFI when stained with the isotype control antibody. The data shows the mean ± SD from three individual experiments. Statistical analysis was performed using the Dunnett’s test for multiple comparisons with V/28/28/3z: * *p* < 0.05 and ***p* < 0.01; and using the Tukey’s test with V/8a/8a/137-3z and V/28/28/Δ137-3z versus V/28/28/137-3z: ^†^
*p* < 0.05 and ^††^
*p* < 0.01; V/8a/8a/Δ137-3z versus V/28/28/Δ137-3z: ^‡^
*p* < 0.05 and ^‡‡^
*p* < 0.01; V/8a/8a/Δ137-3z versus V/8a/8a/137-3z: ^|^
*p* < 0.05 and ^‖^
*p* < 0.01.

**Figure 2 ijms-22-02476-f002:**
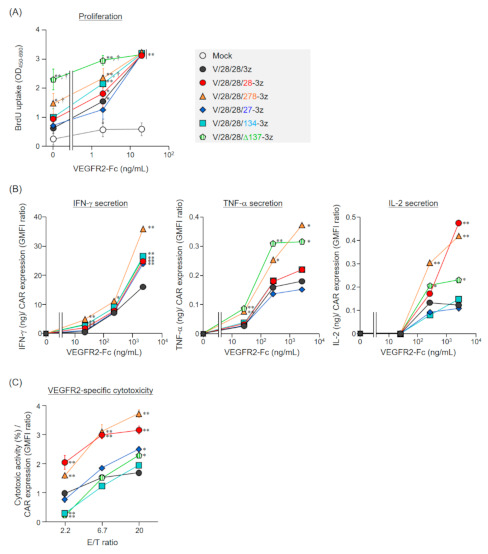
Functional characteristics of mouse T cells expressing second-generation CARs. (**A** and **B**) CAR-T cells cultured for 4 days after Rv transduction, were stimulated with VEGFR2-Fc (0−2000 ng/mL) for 24 h. (**A**) Proliferative activity of STD-modified CAR-T cells following VEGFR2-stimulation. Proliferative activity was analyzed by BrdU uptake ELISA. The data are shown as the mean ± SD of triplicates, and are representative of two independent experiments. Statistical analysis was performed using the Dunnett’s test for multiple comparisons with Mock T cells: * *p* < 0.05 and ** *p* < 0.01; and using the Tukey’s test with second-generation CAR-T cells versus first-generation CAR-T cells (V/28/28/3z): ^†^
*p* < 0.05. (**B**) Relationship between CAR expression level and cytokine-producing ability upon stimulation with mouse VEGFR2-Fc in each of the CAR-T cells. The secretion amount of IFN-γ, TNF-α, or IL-2 from CAR-T cells after 24 h stimulation was determined by ELISA. The data are shown as the mean ± SD of triplicates, and are representative of two independent experiments. Statistical analysis was performed using the Dunnett’s test for multiple comparisons with V/28/28/3z: * *p* < 0.05 and ** *p* < 0.01. (**C**) Relationship between cytotoxicity at indicated E/T ratio and CAR expression level in each of the CAR-T cells. CAR-T cells were cultured for 4 days after Rv-transduction, and then co-cultured with EL4 cells and VEGFR2^+^ EL4 cells at an indicated E/T ratio for 18 h. The numbers of EL4 cells and VEGFR2^+^ EL4 cells in the wells were evaluated by flow cytometry. The cytotoxic activity against VEGFR2^+^ EL4 cells was calculated from the ratio between the numbers of VEGFR2^+^ EL4 cells and EL4 cells. The data shows the mean ± SD of triplicates and are representative of three individual experiments. Statistical analysis was performed using the Dunnett’s test for multiple comparisons with V/28/28/3z: * *p* < 0.05 and ** *p* < 0.01.

**Figure 3 ijms-22-02476-f003:**
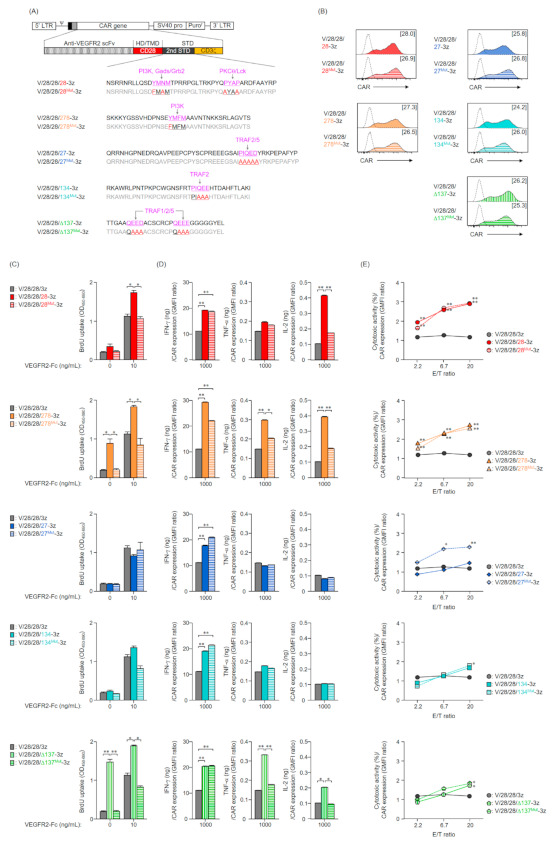
Functional characteristics of mouse T cells expressing second-generation CARs that cannot input the 2nd STD signals. (**A**) Amino acid sequence of STDs derived from costimulatory molecules in second-generation CARs. Adaptor molecules that bind to the 2nd STD in second-generation CAR and their binding regions are shown in magenta. Mutation sites in CARs deficient in the 2nd STD signal input are shown in red. (**B**) CAR expression on T cells after Rv transduction (day 0) was analyzed by flow cytometry using anti-HA-tag (solid color histograms) or isotype control antibodies (dashed white histograms). The expression level of each CAR shown in brackets was calculated from the ratio of GMFI when stained with the anti-HA-tag mAb to GMFI when stained with the isotype control antibody. The data are representative of at least two independent experiments. (C and D) CAR-T cells cultured for 4 days after Rv transduction were stimulated with VEGFR2-Fc (10 or 1000 ng/mL) for 24 h. (**C**) Proliferative activity of CAR-T cells deficient in the 2nd STD signal input following VEGFR2-stimulation. Proliferative activity was analyzed by BrdU uptake ELISA. The data are shown as the mean ± SD of triplicates, and are representative of two independent experiments. Statistical analysis was performed using the Tukey’s test: * *p* < 0.05 and ** *p* < 0.01. (**D**) Relationship between cytokine-producing ability upon stimulation with VEGFR2-Fc 1000 ng/mL and CAR expression level in each of the CAR-T cells. The amount of IFN-γ, TNF-α, or IL-2 secreted by CAR-T cells after 24 h stimulation was determined by ELISA. The data are shown as the mean ± SD of triplicates, and are representative of two independent experiments. Statistical analysis was performed using the Tukey’s test: * *p* < 0.05 and ** *p* < 0.01. (**E**) Relationship between cytotoxicity at indicated E/T ratio and CAR expression level in each of the CAR-T cells. CAR-T cells were cultured for 4 days after Rv transduction, and then were co-cultured with EL4 cells and VEGFR2^+^ EL4 cells at an indicated E/T ratio for 18 h. The numbers of EL4 cells and VEGFR2^+^ EL4 cells in the wells were evaluated by flow cytometry. The cytotoxic activity against VEGFR2^+^ EL4 cells was calculated from the ratio between the numbers of VEGFR2^+^ EL4 cells to that of EL4 cells. The data are shown the mean ± SD of triplicates and are representative of three individual experiments. Statistical analysis was performed using the Tukey’s test with second-generation CARs versus first-generation CAR (V/28/28/3z): * *p* < 0.05 and ** *p* < 0.01.

**Figure 4 ijms-22-02476-f004:**
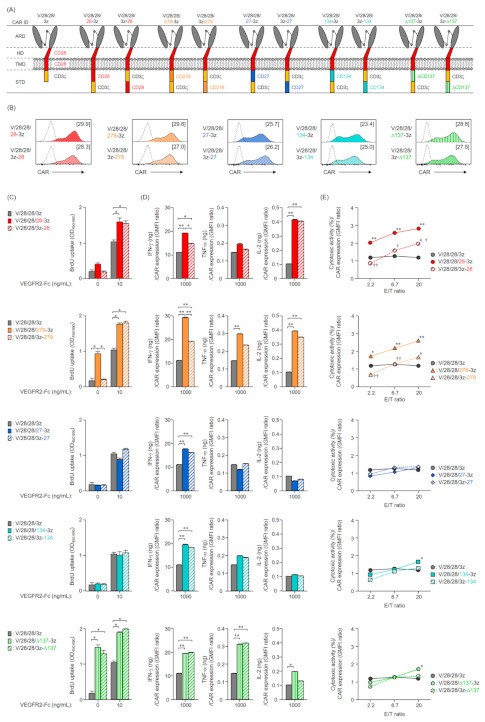
Functional characteristics of mouse T cells expressing second-generation CARs with reordered STDs. (**A**) The illustration of second-generation CARs with the CD3ζ-STD and 2nd STD order reordered. (**B**) CAR expression on T cells after Rv transduction (day 0) was analyzed by flow cytometry using anti-HA-tag (solid color histograms) or isotype control antibodies (dashed white histograms). The expression level of each CAR shown in brackets was calculated from the ratio of GMFI when stained with the anti-HA-tag mAb to GMFI when stained with the isotype control antibody. The data are representative of at least two independent experiments. (**C,D**) CAR-T cells cultured for 4 days after Rv transduction were stimulated with VEGFR2-Fc (10 or 1000 ng/mL) for 24 h. (**C**) Proliferation activity of CAR-T cells with reordered STDs following VEGFR2-stimulation. Proliferative activity was analyzed by BrdU uptake ELISA. The data are shown as the mean ± SD of triplicates, and are representative of two independent experiments. Statistical analysis was performed using the Tukey’s test: * *p* < 0.05 and ** *p* < 0.01. (**D**) The relationship between cytokine-producing ability upon stimulation with VEGFR2-Fc 1000 ng/mL and CAR expression level in each of the CAR-T cells. The amount of IFN-γ, TNF-α, or IL-2 secreted by CAR-T cells after 24 h stimulation was determined by ELISA. The data are shown as the mean ± SD of triplicates, and are representative of two independent experiments. Statistical analysis was performed using the Tukey’s test: * *p* < 0.05 and ** *p* < 0.01. (**E**) The relationship between cytotoxicity at indicated E/T ratio and CAR expression level in each of the CAR-T cells. CAR-T cells were cultured for 4 days after Rv transduction, and then were co-cultured with EL4 cells and VEGFR2^+^ EL4 cells at an indicated E/T ratio for 18 h. Then, the numbers of EL4 cells and VEGFR2^+^ EL4 cells in the wells were evaluated by flow cytometry. The cytotoxic activity against VEGFR2^+^ EL4 cells was calculated from the ratio between the numbers of VEGFR2^+^ EL4 cells to that of EL4 cells. The data are shown the mean ± SD of triplicate and are representative of two individual experiments. Statistical analysis was performed using the Tukey’s test with second-generation CARs versus first-generation CAR (V/28/28/3z): * *p* < 0.05 and ** *p* < 0.01; second-generation CARs with reordered STDs versus unmodified second-generation CARs: ^†^
*p* < 0.05 and ^††^
*p* < 0.01.

**Figure 5 ijms-22-02476-f005:**
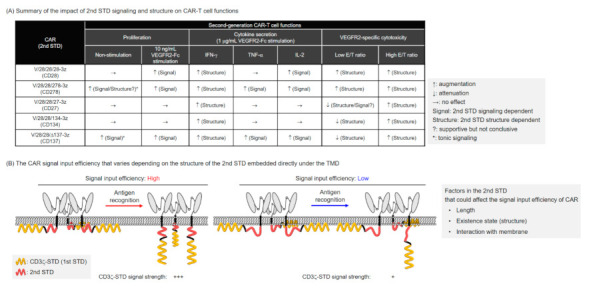
The effects of 2nd STD signaling and structure on the functions of second-generation CAR-T cells. (**A**) The table shows the effect of second-generation CARs inserted directly below the TMD with various co-stimulatory molecule-derived 2nd STD on T cell functions. Arrows indicate increases or decreases in T cell functions when compared to first-generation CAR, and in parentheses, whether these increases or decreases are due to the signal or structure of the 2nd STD. Question marks are assigned if they cannot be definitively attributed to the structure or signal of the 2nd STD. (**B**) The illustration shows that the signal input efficiency of CAR is altered by the 2nd STD structure revealed by this study. Based on the structure of the second-generation CARs used in this study, we speculated that the length and existence state (structure) of the STD and its interaction with the cell membrane may be factors that define the CAR signal input efficiency. The “+” in the figure indicates the strength (input amount) of the CD3ζ−STD signal.

**Table 1 ijms-22-02476-t001:** Gene sequences of the immune molecules used as CAR components in this study.

Components		Gene Sequences
CD3ζ	STD (AA52-164)	AGAGCAAAAT TCAGCAGGAG TGCAGAGACT GCTGCCAACC TGCAGGACCC CAACCAGCTC TACAATGAGC TCAATCTAGG GCGAAGAGAG GAATATGACG TCTTGGAGAA GAAGCGGGCT CGGGATCCAG AGATGGGAGG CAAACAGCAG AGGAGGAGGA ACCCCCAGGA AGGCGTATAC AATGCACTGC AGAAAGACAA GATGGCAGAA GCCTACAGTG AGATCGGCAC AAAAGGCGAG AGGCGGAGAG GCAAGGGGCA CGATGGCCTT TACCAGGGTC TCAGCACTGC CACCAAGGAC ACCTATGATG CCCTGCATAT GCAGACCCTG GCCCCTCGC
CD8α	HD/TMD (AA131-217)	GTCATCAGCA ACTCGGTGAT GTACTTCAGT TCTGTCGTGC CAGTCCTTCA GAAAGTGAAC TCTACTACTA CCAAGCCAGT GCTGCGAACT CCCTCACCTG TGCACCCTAC CGGGACATCT CAGCCCCAGA GACCAGAAGA TTGTCGGCCC CGTGGCTCAG TGAAGGGGAC CGGATTGGAC TTCGCCTGTG ATATTTACAT CTGGGCACCC TTGGCCGGAA TCTGCGTGGC CCTTCTGCTG TCCTTGATCA TCACTCTCAT C
CD28	HD (AA115-177)	ATTGAGTTCA TGTACCCTCC GCCTTACCTA GACAACGAGA GGAGCAATGG AACTATTATT CACATAAAAG AGAAACATCT TTGTCATACT CAGTCATCTC CTAAGCTGTT TTGGGCACTG GTCGTGGTTG CTGGAGTCCT GTTTTGTTAT GGCTTGCTAG TGACAGTGGC TCTTTGTGTG ATCTGGACA
	STD (AA178-218)	AATAGTAGAA GGAACAGACT CCTTCAAAGT GACTACATGA ACATGACTCC CCGGAGGCCT GGGCTCACTC GAAAGCCTTA CCAGCCCTAC GCCCCTGCCA GAGACTTTGC AGCGTACCGC CCC
	STD (AA178-218)^Mut^	AATAGTAGAA GGAACAGACT CCTTCAAAGT GACTTCATGG CCATGACTCC CCGGAGGCCT GGGCTCACTC GAAAGCCTTA CCAGGCCTAC GCCGCTGCCA GAGACTTTGC AGCGTACCGC CCC
	STD (AA178-218)^PI3K-Mut^	AATAGTAGAA GGAACAGACT CCTTCAAAGT GACTTCATGG CCATGACTCC CCGGAGGCCT GGGCTCACTC GAAAGCCTTA CCAGCCCTAC GCCCCTGCCA GAGACTTTGC AGCGTACCGC CCC
	STD (AA178-218)^Lck-Mut^	AATAGTAGAA GGAACAGACT CCTTCAAAGT GACTACATGA ACATGACTCC CCGGAGGCCT GGGCTCACTC GAAAGCCTTA CCAGGCCTAC GCCGCTGCCA GAGACTTTGC AGCGTACCGC CCC
CD278	STD (AA166-200)	TCAAAAAAGA AATACGGATC CAGTGTGCAT GACCCTAATA GTGAATACAT GTTCATGGCG GCAGTCAACA CAAACAAAAA GTCTAGACTT GCAGGTGTGA CCTCA
	STD (AA166-200)^Mut^	TCAAAAAAGA AATACGGATC CAGTGTGCAT GACCCTAATA GTGAATTCAT GTTCATGGCG GCAGTCAACA CAAACAAAAA GTCTAGACTT GCAGGTGTGA CCTCA
CD27	STD (AA204-250)	CAAAGAAGAA ACCACGGGCC AAATGAAGAC CGGCAGGCAG TGCCTGAAGA GCCTTGTCCT TACAGCTGCC CCAGGGAAGA GGAGGGCAGT GCTATCCCTA TCCAGGAGGA CTACCGGAAA CCCGAGCCTG CTTTCTACCC T
	STD (AA204-250)^Mut^	CAAAGAAGAA ACCACGGGCC AAATGAAGAC CGGCAGGCAG TGCCTGAAGA GCCTTGTCCT TACAGCTGCC CCAGGGAAGA GGAGGGCAGT GCTATCGCTG CCGCGGCGGC CTACCGGAAA CCCGAGCCTG CTTTCTACCC T
CD134	STD (AA237-272)	CGGAAGGCTT GGAGATTGCC TAACACTCCC AAACCTTGTT GGGGAAACAG CTTCAGGACC CCGATCCAGG AGGAACACAC AGACGCACAC TTTACTCTGG CCAAGATC
	STD (AA237-272)^Mut^	CGGAAGGCTT GGAGATTGCC TAACACTCCC AAACCTTGTT GGGGAAACAG CTTCAGGACC CCGATCGCGG CGGCCCACAC AGACGCACAC TTTACTCTGG CCAAGATC
CD137	STD (AA209-256)	TCTGTGCTCA AATGGATCAG GAAAAAATTC CCCCACATAT TCAAGCAACC ATTTAAGAAG ACCACTGGAG CAGCTCAAGA GGAAGATGCT TGTAGCTGCC GATGTCCACA GGAAGAAGAA GGAGGAGGAG GAGGCTATGA GCTGT
	STD Δ20 (AA229-256)	ACCACTGGAG CAGCTCAAGC GGCGGCCGCT TGTAGCTGCC GATGTCCACA GGAAGAAGAA GGAGGAGGAG GAGGCTATGA GCTGT
	STD Δ20 (AA229-256)^Mut^	ACCACTGGAG CAGCTCAAGA GGAAGATGCT TGTAGCTGCC GATGTCCACA GGCGGCGGCC GGAGGAGGAG GAGGCTATGA GCTGT

**Table 2 ijms-22-02476-t002:** Amino acid sequences of the immune molecules used as CAR components in this study.

Components		Amino Acid Sequences
CD3ζ	STD (AA52-164)	RAKFSRSAET AANLQDPNQL YNELNLGRRE EYDVLEKKRA RDPEMGGKQQ RRRNPQEGVY NALQKDKMAE AYSEIGTKGE RRRGKGHDGL YQGLSTATKD TYDALHMQTL APR
CD8α	HD/TMD (AA131-217)	VISNSVMYFS SVVPVLQKVN STTTKPVLRT PSPVHPTGTS QPQRPEDCRP RGSVKGTGLD FACDIYIWAP LAGICVALLL SLIITLI
CD28	HD (AA115-177)	IEFMYPPPYL DNERSNGTII HIKEKHLCHT QSSPKLFWAL VVVAGVLFCY GLLVTVALCV IWT
	STD (AA178-218)	NSRRNRLLQS DYMNMTPRRP GLTRKPYQPY APARDFAAYR P
	STD (AA178-218)^Mut^	NSRRNRLLQS DFMAMTPRRP GLTRKPYQAY AAARDFAAYR P
	STD (AA178-218)^PI3K-Mut^	NSRRNRLLQS DFMAMTPRRP GLTRKPYQPY APARDFAAYR P
	STD (AA178-218)^Lck-Mut^	NSRRNRLLQS DYMNMTPRRP GLTRKPYQAY AAARDFAAYR P
CD278	STD (AA166-200)	SKKKYGSSVH DPNSEYMFMA AVNTNKKSRL AGVTS
	STD (AA166-200)^Mut^	SKKKYGSSVH DPNSEFMFMA AVNTNKKSRL AGVTS
CD27	STD (AA204-250)	QRRNHGPNED RQAVPEEPCP YSCPREEEGS AIPIQEDYRK PEPAFYP
	STD (AA204-250)^Mut^	QRRNHGPNED RQAVPEEPCP YSCPREEEGS AIAAAAAYRK PEPAFYP
CD134	STD (AA237-272)	RKAWRLPNTP KPCWGNSFRT PIQEEHTDAH FTLAKI
	STD (AA237-272)^Mut^	RKAWRLPNTP KPCWGNSFRT PIAAAHTDAH FTLAKI
CD137	STD (AA209-256)	SVLKWIRKKF PHIFKQPFKK TTGAAQEEDA CSCRCPQEEE GGGGGYEL
	STD Δ20 (AA229-256)	TTGAAQEEDA CSCRCPQEEE GGGGGYEL
	STD Δ20 (AA229-256)^Mut^	TTGAAQAAAA CSCRCPQAAA GGGGGYEL

## Data Availability

The data presented in this study are available on request from the corresponding author.

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
