# Peer review of "Structure of the Signal Transduction Domain in Second-Generation CAR Regulates the Input Efficiency of CAR Signals"

_ijms, 2021, doi:10.3390/ijms22052476_

Round 1
Reviewer 1 Report
The authors investigate the impact of the structure and signaling via the 2nd STD on CAR T cell proliferation, cytokine secretion and cytotoxic activity. Through extensive and laborious construction of CAR constructs expressing VEGFR2 scFv in a 1st generation CAR with 2nd STD derived from members of the CD28 or TNFRSF families, the authors show that both signaling via the 2nd STD and the structural changes brought in the CAR due to insertion of the 2nd STD at the appropriate position are both critical for effective CAR T function. While tonic signaling is observed only when the 2nd STD is derived from CD278 or CD137, CAR with 2nd STD from CD28 and CD278 have the highest cytotoxic potential that is dependent on the order of domains, but independent of the signaling motif. Interestingly, tonic signaling via CD137 STD-CAR is independent of the structure. The authors also elegantly show that different cytokine secretion are regulated differently. While TNFa and IL-2 secretion are dependent on the signaling motif, IFNg secretion is dependent on the presence of 2nd STD only in between TMD and CD3z STD.
The experiments are well executed and the methods well laid out with helpful illustrations. The paper makes a significant contribution to the field and will benefit by addressing a few additional questions in the future.
1) How effective are these CAR in vivo in a VEGFR2-expressing tumor model?
2) How do the reordering and signaling motif mutants influence CAR function in vivo?
Author Response
Reply to the Review Report (Reviewer 1):
We appreciate you taking the time to offer us your comments related to the paper. We are very grateful that you have a good understanding of the new findings on STD that we wanted to convey in this paper.
As you commented, we need to clarify the relationship between the findings of the present study and the in vivo efficacy and behavior of second-generation CAR-T cells. In particular, we are highly interested in analyzing the effect of different STD order of second generation CARs with CD28-STD or CD278-STD on the functionality of these CAR-T cells in vivo. We are currently constructing mouse models of hematological and solid cancers to evaluate the in vivo function of CAR-T cells so that we can report these new findings in the near future. Please wait for further updates.
We have revised the manuscript based on the comments from other reviewers. The changes in the text are underlined and highlighted in yellow.
Reviewer 2 Report
General comment:
This work by Fujiwara, Okada and colleagues aimed at controlling the input efficiency of CAR signals through different signal transduction domains. The manuscript could benefit from more precise language formulations at times, nonetheless this is minor and certainly addressable. As it stands this manuscript should undergo major revision and precise clarification of some datasets, both in figures and text, that it can be considered for publication. The manuscript does, nonetheless, not provide any insight of these effects in vivo, thus being short to show if the factors herein analyzed actually have a biological impact on tumor clearance.
Specific comments:
Line 28. Convenient method for creating specific anti-tumor antigen-specificity in T cells. There are still limitations to mass production of CAR T cells. See, Rafiq et al. Nat. Rev. Clinical Oncology 2019.
Line 51: “However, the bulk of CAR research has been focused on finding target antigens of promise for treating refractory cancers, and on evaluating the efficacy of CAR-T cell therapy targeting these antigens [15, 16].” Please consider recent reviews such as Rafiq et al. Nat. Rev. Clinical Oncology 2019 or Lim et al. Cell 2017 which demonstrate that there is actually a vast amount of literature covering several limitations to CAR T cell therapy beyond antigen recognition.
Line 53: “Only limited knowledge has been accumulated about the structural design of CARs to fine-tune CAR-T cell functions.” I certainly agree with this knowledge limitation, but currently there exist at list 2 studies that looked into the fine-tuning of these therapies and these are not mentioned neither here, nor later in the discussion despite extremely relevant for the topic herein presented: Feucht et al. Nat. Med. 2019 and indirectly relevant (as it looks into the integration site rather than the specific design of the CAR) Eyquem et al. Nature 2017.
Figure 1 B and D: A percentage of transduced cells would also be quite interesting at this point and the information provided by it would be complementary to the MFI presented. Perhaps complemented with percentage of live and dead cells.
Figure 1 B and D: Although the mRNA level is comparable across time between the different constructs, the GMFI ratio shows clear drops in CAR expression levels over time. Was it just a reduction in GMFI ratio or was it also accompanied by a reduction in transduction efficiency (meaning, actual reduction of transduced cells in the condition)?
Line 156: “Taken together, these results suggested that the additional insertion of the 2nd STD after the TMD affects the biosynthesis of CAR proteins and the efficiency of their transport to the membrane.” à There is no data supporting this general statement, this statement should be restricted to CD137. The experiments show that the insertion of CD28, CD278, CD27, CD134 did not significantly impact the expression levels of CAR. Only CD137 did reduce expression level. Despite suggestive thereof, no data is provided showing that indeed there is impaired biosynthesis or impaired transport efficiency to the membrane.
Figure 2A and B: It would be interesting and perhaps a little more elucidating to show proliferation of these cells in a co-culture with antigen positive and antigen negative cells over serial time-points. Furthermore, the statement in line 169 does not seem congruent with figure 2B, since those constructs deemed to be tonic signaling show no increase in cytokines without stimulation. Should the proliferation data not go hand in hand with cytokine to support the evidence of tonic signaling?
Figure 2B and C: all results are presented normalized to the GMFI ratio, at which day was this GMFI ratio measured and assessed, day 4? Possible that this normalization is making the results rather artificial and harder to interpret without any added benefit? How would this data look like without normalization?
Figure 2C: again, uncommon to present cytotoxicity normalized in such a manner as it removes the possibility to understand which percentage of target cells were actually lysed.
Line 211 to 213: there is no data in this manuscript that support the claim that these CARs prevented T cell exhaustion.
2.3. Expression and function in mouse T cells of CAR mutants deficient in the co-stimulatory 219 signal input motif
Figure 3B and 3C,D and E: The numbers in 3B, in the right upper corner of each plot correspond to the percentage of cells expressing the construct or GMFI? If percentage it certainly seems much higher than that from looking at the plot and they are all extremely comparable, so in the following figures again, is there really the need to normalization to GMFI? Furthermore, in the figure legend it is described that expression of CAR after retroviral transduction was checked on day 0 and the assay (which is normalized to GMFI) was performed on day 4 after retroviral transduction. Nevertheless, on Figure 1 we can observe that there is a significant drop on all conditions from day 0 to day 4. Please comment.
Figure 3E: In this cytotoxicity it seems that V/28/28/3z has little to no activity, in my opinion due to the normalization step, would it be possible to present the results in percentage of lysed cells? The question goes into the direction of having a clear understanding from the graph if the modifications in STD improve cytotoxicity from xx% t yy% killed cells.
Line 278: “A series of analyses of CAR [V/28/28/27-3z]-T cells showed that CD27-STD signaling did not enhance the functions of CAR-T cells, suggesting that the mutation of the signal input motif in CD27-STD enhanced the input efficiency of CD3z-STD signaling by altering the structure of the STD.” This sentence is not well fundamented, lacking on one hand the figures it refers to, and if referring to figure 3, for example in fig 3D, both 27 and 27mut have improved gamma secretion and are not significant between each other in any parameter. Looking into fig 3E there is indeed a change in cytotoxicity, but that is not sufficient to justify the conclusion of “enhanced input efficiency of CDzeta-STD signaling by altering the structure of the STD.”.
Line 282: “Collectively, these results indicated that the enhanced secretion of TNF-a and IL-2, and antigen-independent cell proliferation by CAR-mediated tonic signaling in second-generation CAR-T cells are brought about by the input of 2nd STD signals.” This sentence lack on one hand support of no antigen stimuli for the cytokine data, would the effects seen still stand without antigen stimuli? Furthermore, although I agree with the authors I think that the 27 and the 134 cases should perhaps be pointed out, since they did not impact at all proliferation without antigen (but the 134mut seems significantly lower in the proliferation with antigen, please comment) or cytokines. Furthermore, at this point I cannot avoid but wondering why 10 ng/ml of VEGFR2-Fc was used for stimulation for the proliferation assay and 1000 ng/ml for the cytokine assay. Perhaps the whole dataset would be more valuable if generated in a setting of coculture with antigen positive or antigen negative cells.
Line 285: “In contrast, the enhancement in IFN-g secretion and cytotoxic activity of the second-generation CAR-T cells was not due to the contribution of the 2nd STD signal, but rather to a change in the signal input efficiency of the 1st STD (CD3z-STD) due to structural changes in the STD of CAR.” Since the construct 278 shows differences between 278 and 278mut (significantly lower), it could be postulated that actually this signaling domain has an impact, unlike the general comment made in this sentence. The authors statement could perhaps be proven if this signaling domain is replaced a different aminoacids amounting to the same length, reconstituting the highest peak of gamma.
Line 325: “This suggested that the second-generation CARs with DCD137-STD have an STD structure that facilitates the formation of TRAF-mediated signalosomes regardless of the insertion position of DCD137-STD.” Not proven at all, this “suggestion” is an ideological jump that would require some experimental control.
Figure 5: More based on a literature review than on the original findings herein presented. This manuscript has no data investigating ITAM phosphorylation and its correlation to the findings herein presented.
Author Response
Reply to the Review Report (Reviewer 2):
We appreciate you taking the time to offer us your comments and insights related to the paper. We found your feedback very constructive and tried to be responsive to your concerns. Our responses are given in a point-by-point manner below. Changes in the manuscript are underlined and highlighted in yellow.
What we do not want you to misunderstand is that the aim of the present study is to closely examine the effect of STD structure on CAR expression and activity to optimize CAR-T cells. Many previous studies have strongly suggested that the functional strength of CAR-T cells depends on the CAR expression level. The CAR expression level would depend not only on the CAR structure but also on the method of CAR gene transfer. Therefore, we thought that the collection of information on the activity of CAR proteins, but not the function of CAR-T cells, could provide useful information to a wide range of CAR-T cell therapy researchers. In this study, we standardized some functional intensities of CAR-T cells with CAR expression levels. With the standardization process, we believe we highlighted that the signal input efficiency of CAR proteins varies depending on the type and position of the 2nd STD. Based on this finding, it will also be important to consider the method of CAR expression (gene transfer method and CAR gene expression level) according to the structure of CAR. The experimental data necessary for the interpretation of the Figure presented in this study has been added as a Supplementary Figure.
As you pointed out, we understand that it is important to analyze the relationship between the findings of this study and the in vivo antitumor effects, but we have not been able to conduct in vivo functional analysis of various second-generation CAR-T cells. We are currently constructing an in vivo evaluation system for hematological and solid tumors, so please wait for further reports on these in vivo analysis data.
Q1;
Line 28. Convenient method for creating specific anti-tumor antigen-specificity in T cells. There are still limitations to mass production of CAR T cells. See, Rafiq et al. Nat. Rev. Clinical Oncology 2019.
A1;
We agree that there are still challenges in the mass preparation of functional CAR-T cells. However, we believe that the transduction of CAR genes into T cells is a method with high potential for the mass preparation of tumor cell-specific T cells, compared to the conventional method of preparing tumor associated antigen-specific T cells using tumor-infiltrating lymphocytes. The text has been revised as follows.
Page 2, lines 26-28;
(Before revision)
Transduction of chimeric antigen receptor (CAR) gene, which is a fusion protein of a tumor antigen-binding molecule and a T cell-activating molecule, into T cells is a convenient method for the mass production of tumor cell-specific cytotoxic T cells (CAR-T cells).
(After revision)
Chimeric antigen receptor (CAR) is a fusion protein of a tumor antigen-binding molecule and a T cell-activating molecule. Transduction of CAR gene into T cells is a convenient method for creating the tumor cell-specific cytotoxic T cells (CAR-T cells).
Q2;
Line 51: “However, the bulk of CAR research has been focused on finding target antigens of promise for treating refractory cancers, and on evaluating the efficacy of CAR-T cell therapy targeting these antigens [15, 16].” Please consider recent reviews such as Rafiq et al. Nat. Rev. Clinical Oncology 2019 or Lim et al. Cell 2017 which demonstrate that there is actually a vast amount of literature covering several limitations to CAR T cell therapy beyond antigen recognition.
A2;
Thank you for your remarks. We have determined that this statement is not appropriate and have revised it as follows.
Page 2, lines 50-54;
(Before revision)
However, the bulk of CAR research has been focused on finding target antigens of promise for treating refractory cancers, and on evaluating the efficacy of CAR-T cell therapy targeting these antigens.
(After revision)
Although a variety of CAR structures have been developed, these CARs are unique to the researchers and there is a lack of studies that have analyzed the linkage between the structure and activity of artificial CAR proteins in detail.
Q3;
Line 53: “Only limited knowledge has been accumulated about the structural design of CARs to fine-tune CAR-T cell functions.” I certainly agree with this knowledge limitation, but currently there exist at list 2 studies that looked into the fine-tuning of these therapies and these are not mentioned neither here, nor later in the discussion despite extremely relevant for the topic herein presented: Feucht et al. Nat. Med. 2019 and indirectly relevant (as it looks into the integration site rather than the specific design of the CAR) Eyquem et al. Nature 2017.
A3;
We recognize that the two references you presented are important for the optimization of CAR-T cells. Reference 1 (Feucht et al.) will contribute to the optimization of CAR intracellular regions together with our present findings, and reference 2 (Eyquem et al.) will be needed to achieve the approach of changing the expression intensity in view of the activity of CAR. In order to present directions for future research, these references have been added to the Discussion rather than the Introduction. (Page17-18, line476-507)
Q4;
Figure 1 B and D: A percentage of transduced cells would also be quite interesting at this point and the information provided by it would be complementary to the MFI presented. Perhaps complemented with percentage of live and dead cells.
A4;
We have established a highly efficient (close to 100% efficiency) method for CAR gene delivery to mouse T cells using retroviral vectors (Rv). The data of CAR expression in CD8 T cells in this study have been added to the Supplementary Figure 1. By reviewing these results, you will be convinced that the expression level of each CAR (GMFI) shown in this study is not the efficiency of CAR transduction but the true efficiency of CAR expression.
In the analysis, we confirmed that there was no obvious abnormality in the percentage of dead cells or T cells depending on the structure of the Rv-transfected CARs.
Q5;
Figure 1 B and D: Although the mRNA level is comparable across time between the different constructs, the GMFI ratio shows clear drops in CAR expression levels over time. Was it just a reduction in GMFI ratio or was it also accompanied by a reduction in transduction efficiency (meaning, actual reduction of transduced cells in the condition)?
A5;
It is interesting to note that the level of CAR expression on the T cell membrane gradually decreased with the passage of culture days. In order to analyze the relationship between CAR structure and CAR-T cell function in detail, we have devised a method for the generation of CAR-T cells as described above, and have selectively cultured transgenic cells using medium with a high concentration of puromycin. That is, we speculate that the reason for the decrease in CAR expression level over time is not a decrease in the percentage of transgenic cells, but a decrease in CAR protein expression in T cells. Although the details are unclear, we believe that these phenomena are caused by a decrease in the amount of protein synthesis in T cells, since T cells are assumed to shift from an activated state to a quiescent state with the passage of culture days.
Q6;
Line 156: “Taken together, these results suggested that the additional insertion of the 2nd STD after the TMD affects the biosynthesis of CAR proteins and the efficiency of their transport to the membrane.” à There is no data supporting this general statement, this statement should be restricted to CD137. The experiments show that the insertion of CD28, CD278, CD27, CD134 did not significantly impact the expression levels of CAR. Only CD137 did reduce expression level. Despite suggestive thereof, no data is provided showing that indeed there is impaired biosynthesis or impaired transport efficiency to the membrane.
A6;
We apologize for the excessive wording. The text has been revised as follows.
Page 5, lines 161-162;
(Before revision)
Taken together, these results suggested that the additional insertion of the 2nd STD after the TMD affects the biosynthesis of CAR proteins and the efficiency of their transport to the membrane.
(After revision)
Taken together, these results suggested that the additional insertion of the 2nd STD after the TMD affects the efficiency of membrane expression of CAR proteins.
Q7;
Figure 2A and B: It would be interesting and perhaps a little more elucidating to show proliferation of these cells in a co-culture with antigen positive and antigen negative cells over serial time-points. Furthermore, the statement in line 169 does not seem congruent with figure 2B, since those constructs deemed to be tonic signaling show no increase in cytokines without stimulation. Should the proliferation data not go hand in hand with cytokine to support the evidence of tonic signaling?
A7;
We have confirmed that all CAR-T cells did not secrete various cytokines in the condition without antigen stimulation (VEGFR2-Fc 0 ng/mL condition). The lack of cytokine secretion despite the antigen-independent proliferative activity in the two CAR-T cells (V/28/28/278-3z and V/28/28/D137-3z) may be due to the difference in sensitivity of the experimental method. As these CAR-T cells exerted antigen-specific effector functions, the tonic signals detected in this study were very weak and may not have a negative effect on T cells. However, we noted that the fact that tonic signals are input by STD structures should be considered as a factor affecting the quality of CAR-T cells and their viability in vivo. Further studies on the relationship between the intensity of tonic signals and the quality and function of CAR-T cells are desirable.
Q8;
Figure 2B and C: all results are presented normalized to the GMFI ratio, at which day was this GMFI ratio measured and assessed, day 4? Possible that this normalization is making the results rather artificial and harder to interpret without any added benefit? How would this data look like without normalization?
Q9;
Figure 2C: again, uncommon to present cytotoxicity normalized in such a manner as it removes the possibility to understand which percentage of target cells were actually lysed.
A8 and A9;
For the functional analysis of all CAR-T cells, CAR-T cells at 4 days after Rv transduction were used. Each data shows the ratio of functional intensity and CAR expression level of the CAR-T cells used. The functional intensity of each CAR-T cell before standardization was added to the Supplementary Figure 2. As mentioned in the beginning, we believe that standardization of CAR expression level is important in order to clarify the activity of CARs.
Q10;
Line 211 to 213: there is no data in this manuscript that support the claim that these CARs prevented T cell exhaustion.
A10;
Thank you for pointing this out. The text has been revised to the following.
Page 7, lines 216-218;
(Before revision)
These results suggest that second-generation CARs with STDs derived from members of the TNFRSF have a high activation threshold compared with first-generation CARs, but can exert strong cytotoxic activity under conditions of a high E/T ratio by preventing T cell exhaustion and secreting large amounts of cytokines.
(After revision)
These results suggested that the CAR ability to induce CD3z-STD signaling-mediated cytotoxic activity was altered by differences in the type of 2nd STD (structure and signal input mechanism).
Q11;
Figure 3B and 3C,D and E: The numbers in 3B, in the right upper corner of each plot correspond to the percentage of cells expressing the construct or GMFI? If percentage it certainly seems much higher than that from looking at the plot and they are all extremely comparable, so in the following figures again, is there really the need to normalization to GMFI? Furthermore, in the figure legend it is described that expression of CAR after retroviral transduction was checked on day 0 and the assay (which is normalized to GMFI) was performed on day 4 after retroviral transduction. Nevertheless, on Figure 1 we can observe that there is a significant drop on all conditions from day 0 to day 4. Please comment
A11;
The number in the upper right corner of the histogram indicates the CAR expression level (GMFI ratio). We apologize for not including it in the figure legend and have added it to all the relevant sections.
The expression levels of all CARs were analyzed every 2 days after Rv transduction, and since the expression efficinecy and loss profile of 2nd STD signal-deficient CARs and STD-ordered modified CARs were similar to those of unmodified CARs, the results immediately after Rv transduction (day 0) are shown as representative. This information has been added as follows.
Page 7, lines 233-236;
The surface expression levels of these mutant CARs were comparable to those of the un-modified second-generation CARs (Figure 3B), and the membrane expression profiles of these CARs were not markedly different (data not shown).
Page 10, lines 307-312;
The expression level in mouse T cells of the CARs with reordered STDs was comparable to that of the unmodified second-generation CARs (Figure 4B), and the expression profiles of these CARs were also not different (data not shown). This result indicated that reordering of the STDs did not affect the CAR expression efficiency.
Q12;
Figure 3E: In this cytotoxicity it seems that V/28/28/3z has little to no activity, in my opinion due to the normalization step, would it be possible to present the results in percentage of lysed cells? The question goes into the direction of having a clear understanding from the graph if the modifications in STD improve cytotoxicity from xx% t yy% killed cells.
A12;
The results of cytotoxic activity have been added to the Supplementary Figure. We confirmed that both CAR-T cells specifically killed VEGFR2-expressing cells. Again, in order to exclude the effect of CAR expression level on the functional strength of CAR-T cells, we standardized the cytotoxic activity of CAR-T cells by each CAR expression level.
Q13;
Line 278: “A series of analyses of CAR [V/28/28/27-3z]-T cells showed that CD27-STD signaling did not enhance the functions of CAR-T cells, suggesting that the mutation of the signal input motif in CD27-STD enhanced the input efficiency of CD3z-STD signaling by altering the structure of the STD.” This sentence is not well fundamented, lacking on one hand the figures it refers to, and if referring to figure 3, for example in fig 3D, both 27 and 27mut have improved gamma secretion and are not significant between each other in any parameter. Looking into fig 3E there is indeed a change in cytotoxicity, but that is not sufficient to justify the conclusion of “enhanced input efficiency of CDzeta-STD signaling by altering the structure of the STD.”.
A13;
As you pointed out, this text has been deleted from the text because it was judged to be inappropriate, and the following text has been added in the Discussion for the discussion of CARs with CD27.
Page 9, lines 484-489;
In this study, we found that second-generation CAR with CD27-STD did not enhance TNF-a secretion by CD27-STD signaling, while the loss of TRAF-binding motifs induced high cytotoxic activity, although the cause of this effect was not clarified in detail. These findings suggest that the TNFRSF-derived STD in the second-generation CAR shows a different structural state from that of endogenous TNFRSF and may interact with molecules other than TRAFs.
Q14;
Line 282: “Collectively, these results indicated that the enhanced secretion of TNF-a and IL-2, and antigen-independent cell proliferation by CAR-mediated tonic signaling in second-generation CAR-T cells are brought about by the input of 2nd STD signals.” This sentence lack on one hand support of no antigen stimuli for the cytokine data, would the effects seen still stand without antigen stimuli? Furthermore, although I agree with the authors I think that the 27 and the 134 cases should perhaps be pointed out, since they did not impact at all proliferation without antigen (but the 134mut seems significantly lower in the proliferation with antigen, please comment) or cytokines. Furthermore, at this point I cannot avoid but wondering why 10 ng/ml of VEGFR2-Fc was used for stimulation for the proliferation assay and 1000 ng/ml for the cytokine assay. Perhaps the whole dataset would be more valuable if generated in a setting of coculture with antigen positive or antigen negative cells.
A14;
To repeat answer 7, we have confirmed that these CAR-T cells do not secrete cytokines in an antigen-independent manner. Therefore, we assumed that the two types of CAR-T cells received a very weak activation signal input through the CAR. We also added the following text on the results of proliferation assay of second generation CARs with CD27-STD or CD134-STD. In addition, we have confirmed that the deletion of the 2nd STD signaling motif in the second-generation CAR with CD134-STD tends to affect the proliferative activity of CAR-T cells, but the difference is not statistically significant.
The aim of the experiment shown in Figure 3 is to clarify whether the function in second generation CAR-T cells affects the signal input of the 2nd STD. Since the sensitivity of the cell proliferation assay and that of the cytokine secretion assay are different, we took into account the results of the experiment in Figure 2 and extracted the conditions under which the functional difference between the second generation CAR-T cells and the first generation CAR-T cells was observed.
The evaluation of CAR-T cell function using antigen-expressing cells that you proposed is an excellent method for assessing the functional strength of CAR-T cells, but it is slightly different from our desired findings. Considering the results of this study, it is necessary to examine the activity of many CAR structures to deeply understand the CAR structure-activity relationship, and a simple and highly reproducible experimental system is desirable to quickly collect useful information on the structural design of CARs. We believe that the experiment using VEGFR2-Fc is an excellent method to properly understand the CAR structure-activity relationship because it allows flexible setting of the antigen density and has excellent reproducibility. Of course, the most important issue is to reflect the findings of this study as the performance of CAR-T cells, so we added the following sentence to Discussion in relation to Q3 (Page 17, line 497-507).
Page 9, lines 272-274;
As in T cells expressing unmodified second-generation CARs, T cells expressing CARs deficient in the 2nd STD signal input motif secreted several cytokines upon antigen stimulation for the first time (Figure 3D).
Page 9, lines 278-280;
The ability of second-generation CARs with CD27-STD or CD134-STD to induce cell proliferation activity and cytokine secretion was not affected by the lack of their signaling motifs.
Q15;
Line 285: “In contrast, the enhancement in IFN-g secretion and cytotoxic activity of the second-generation CAR-T cells was not due to the contribution of the 2nd STD signal, but rather to a change in the signal input efficiency of the 1st STD (CD3z-STD) due to structural changes in the STD of CAR.” Since the construct 278 shows differences between 278 and 278mut (significantly lower), it could be postulated that actually this signaling domain has an impact, unlike the general comment made in this sentence. The authors statement could perhaps be proven if this signaling domain is replaced a different aminoacids amounting to the same length, reconstituting the highest peak of gamma.
A15;
The sentence you pointed out was indeed inappropriate, and we have revised it as shown in the text below. As you suggested, we are planning to conduct an analysis using a new STD-modified CAR structure to examine whether the increased secretion of IFN-g in the second-generation CAR is affected by the length and structure of the 2nd STD. Please look forward to further reports.
Page 7, lines 216-218;
(Before revision)
In contrast, the enhancement in IFN-g secretion and cytotoxic activity of the second-generation CAR-T cells was not due to the contribution of the 2nd STD signal, but rather to a change in the signal input efficiency of the 1st STD (CD3z-STD) due to structural changes in the STD of CAR.
(After revision)
In contrast, the enhancement in IFN-g secretion and cytotoxic activity of the second-generation CAR-T cells was a small contribution of the 2nd STD signal.
Q16;
Line 325: “This suggested that the second-generation CARs with DCD137-STD have an STD structure that facilitates the formation of TRAF-mediated signalosomes regardless of the insertion position of DCD137-STD.” Not proven at all, this “suggestion” is an ideological jump that would require some experimental control.
A16;
As you pointed out, the sentence has been deleted due to its logical jump. To deepen our knowledge of tonic signals in second-generation CARs with CD137-STD, we have revised or added the following sentence to Discussion.
Page 15, lines 432-441;
Previous studies using human second-generation CAR with CD137-STD reported that constant input of CD137-STD-mediated TRAF2 signaling led to antigen-independent pro-liferation of CAR-T cells [25].
Page 15, lines 439-441;
This may suggest that the second-generation CARs with DCD137-STD have an STD structure that facilitates the formation of TRAF2-mediated signalosomes regardless of the insertion position of DCD137-STD.
Q17;
Figure 5: More based on a literature review than on the original findings herein presented. This manuscript has no data investigating ITAM phosphorylation and its correlation to the findings herein presented.
A17;
Based on the findings of this study, we have revised the figure to summarize the factors that determine the signal input efficiency of CARs. The previous figure has been moved to the Supplementary Figure 7 for reference.
Again, thank you for giving us the opportunity to strengthen our manuscript with your valuable comments and queries. We have worked hard to incorporate your feedback and hope that these revisions persuade you to accept our submission.
Reviewer 3 Report
Optimising CAR domain sequence and configuration is a high-parameter combinatorial problem that has not, until recently, been done either thoroughly or systematically in the field. As the authors of this study correctly point out, comparisons among the most-studied 2nd-gen CAR configurations from different groups or companies are complicated by differences in multiple domains. In the CD19 field, for example, most CARs use the same antigen-binding domain and zeta signalling tail but differ in their choice of hinge, TM and co-stimulatory sequences. Disparate functional profiles thus cannot be precisely ascribed to one particular domain or sequence when the constructs being compared have multiple differences.
In this manuscript, Fujiwara et al follow on from their systematic study of hinge-TMD configurations (Fujiwara et al, Cells 2020) by taking a similarly systematic approach to varying co-stimulatory sequences and configurations analysed in mouse T cells using standard in vitro functional assays. This kind of study is very important for the field and the results presented here will be helpful to many researchers making decisions about receptor design. The data presented are fundamentally sound and the methods are appropriate. However, there are some significant issues with data presentation and interpretation that need to be addressed:
Major issues:
- All of the expression, cytokine production and cytotoxicity data in the paper are presented as derivatised ratios of one kind or another. While this is not an unreasonable way to present data when relative activities are most important, this loses a great deal of useful information that is in the primary data. Expression levels are shown as a ratio of MFI between specific and isotype control staining – this does not inform on the distribution of expression levels in the population and the original histograms for Figure 1 B/D should be shown (at least in a supplementary figure) as they are in Figure 3B. Likewise, cytokine quantitation and cytotoxicity in all figures should be shown in their original form without normalisation by expression levels (which is itself an abbreviated ratiometric representation of MFI). Again, this could be added as supplementary data but is required so that readers familiar with normal activity ranges can judge the quality of the assay.
- In the results describing Figure 1 the authors ascribed the low expression of V/28/28/137-3z to a combination of CD28 TMD and the 20-amino-acid membrane-proximal stretch containing many basic residues. But they later point out (lines 404/405) that their CD137 tail sequence contains 3 additional residues (SVL) close to the TMD and conclude that this sequence affected expression. They have not tested this assertion experimentally and should clarify in the text that this is conjecture. In addition, the recommendation to use structure prediction software that follows (lines 407-410) is of dubious value, and this reviewer suggests that a topology predictor such as TMHMM would be much more informative.
- Line 157: the authors have ascribed differing surface levels to “biosynthesis” and/or “transport to the membrane” but it could just as easily be differing internalisation rates, stability in the lipid bilayer, sorting into membrane sub-compartments or even total protein synthesis. Nailing this down is probably outside the scope of this study, but western blots measuring total protein synthesis for all constructs would narrow it down. This is a more useful measure than mRNA levels and would add significantly to the analysis.
- Figure 5 contains a great deal of speculative implication about tail conformations and interactions with the membrane that are not measured in this study. B(ii) implies that the CD28 tail disrupts interactions with the membrane, but CD28 has been shown to behave very much like CD3 in its membrane interactions (Dobbins et al, Sci Signal 2016) so one might expect their effects to be additive in suppressing spontaneous signalling. In addition, B(iii) is not called out in the text at all and is not very informative. The authors should reconsider the point of this summary figure because it seems to suggest that all differences among co-stimulatory domains are due to effects on interactions with the membrane which, again, was not measured here.
- The authors should include a discussion of their results in the context of other studies that did examine position effects when combining activation and co-stimulatory sequences, for example Maher et al Nat Biotech 2002; Finney et al JI 1998.
Minor issues:
- Line 82: The abbreviation V/28/28/3z is first used here, but it isn’t clear here what the “V” stands for. The antigen specificity (VEGFR) should be described here for clarity.
- Line 212: Exhaustion was not measured in this study, and this comment is a bit speculative. Cytotoxicity assay is only 18 hours, and not long enough to demonstrate exhaustion.
- Line 252-254: First sentence states “none of the second gen CAR-T cells showed antigen-independent proliferation”, however V/28/28/278-3z and V/28/28/137-3z do show significant proliferation in Figure 3C. Please clarify.
- Line 353-354: Position of 2nd STD only effects cytotoxicity and IFNg secretion of CD28 family STD CARs, but not TNFRSF STD CARs. This conclusion cannot be generalised to all co-stimulatory sequences. Same again in the discussion, lines 473-475
- Line 427-428: I do not agree with this statement. There is wide appreciation that signalling from the native TCR is much more nuanced and regulated than signalling from CARs and there is little evidence that force or conformation play a role in translating antigen binding into ITAM phosphorylation.
- Line 463-465: These reagents are widely available. If the analysis in question is beyond the scope of this study the authors should simply highlight a biochemical analysis of signalling pathways as a useful further analysis.
- Line 470-471: This statement only applies to the CD137-derived STD and should not be generalised.
Author Response
Reply to the Review Report (Reviewer 3):
We appreciate you taking the time to offer us your comments related to the paper. We are very grateful that you have a good understanding of the new findings on HD/TMD that we wanted to convey in this paper. Our responses are given in a point-by-point manner below. Changes in the manuscript are underlined and highlighted in yellow.
Q1;
All of the expression, cytokine production and cytotoxicity data in the paper are presented as derivatised ratios of one kind or another. While this is not an unreasonable way to present data when relative activities are most important, this loses a great deal of useful information that is in the primary data. Expression levels are shown as a ratio of MFI between specific and isotype control staining – this does not inform on the distribution of expression levels in the population and the original histograms for Figure 1 B/D should be shown (at least in a supplementary figure) as they are in Figure 3B. Likewise, cytokine quantitation and cytotoxicity in all figures should be shown in their original form without normalisation by expression levels (which is itself an abbreviated ratiometric representation of MFI). Again, this could be added as supplementary data but is required so that readers familiar with normal activity ranges can judge the quality of the assay.
A1;
The primary data for CAR expression, cytokine secretion, and cytotoxic activity have been added to the Supplementary Figure 1-4.
Q2;
In the results describing Figure 1 the authors ascribed the low expression of V/28/28/137-3z to a combination of CD28 TMD and the 20-amino-acid membrane-proximal stretch containing many basic residues. But they later point out (lines 404/405) that their CD137 tail sequence contains 3 additional residues (SVL) close to the TMD and conclude that this sequence affected expression. They have not tested this assertion experimentally and should clarify in the text that this is conjecture. In addition, the recommendation to use structure prediction software that follows (lines 407-410) is of dubious value, and this reviewer suggests that a topology predictor such as TMHMM would be much more informative.
A2;
In order to properly discuss our hypothesis that the presence of non-polar amino acids affected CAR expression, we have added mouse and human CD137-STD sequences to the Supplementary Table 1 and revised the text below. In addition, we revised the Topology predictor to TMHMM as you suggested.
Page 15, lines 419-423;
Compared to the human CD137-STD, the mouse CD137-STD used in this study contains three extra amino acids (Ser-Val-Leu) at the N-terminal end adjoining the TMD (Supplementary Table 1). Given that there are no reports that human CARs with CD137-STD are refractory to expression [Ref], the presence of these non-polar amino acids may be a factor that reduces the expression efficiency of V/28/28/137-3z.
Q3;
Line 157: the authors have ascribed differing surface levels to “biosynthesis” and/or “transport to the membrane” but it could just as easily be differing internalisation rates, stability in the lipid bilayer, sorting into membrane sub-compartments or even total protein synthesis. Nailing this down is probably outside the scope of this study, but western blots measuring total protein synthesis for all constructs would narrow it down. This is a more useful measure than mRNA levels and would add significantly to the analysis.
A3;
As you pointed out, the low level of membrane expression in some CARs could be due to various factors, which we have not been able to identify in this study. In the future, we would like to clarify the CAR structural elements that affect the expression mechanism of CAR proteins by western blotting analysis.The relevant text has been revised as follows.
Page 5, lines 161-162;
(Before revision)
these results suggested that the additional insertion of the 2nd STD after the TMD affects the biosynthesis of CAR proteins and the efficiency of their transport to the membrane.
(After revision)
these results suggested that the additional insertion of the 2nd STD after the TMD affects the surface expression level of CAR proteins.
Q4;
Figure 5 contains a great deal of speculative implication about tail conformations and interactions with the membrane that are not measured in this study. B(ii) implies that the CD28 tail disrupts interactions with the membrane, but CD28 has been shown to behave very much like CD3 in its membrane interactions (Dobbins et al, Sci Signal 2016) so one might expect their effects to be additive in suppressing spontaneous signalling. In addition, B(iii) is not called out in the text at all and is not very informative. The authors should reconsider the point of this summary figure because it seems to suggest that all differences among co-stimulatory domains are due to effects on interactions with the membrane which, again, was not measured here.
A4;
Based on the findings of this study, we have revised the figure to summarize the factors that determine the signal input efficiency of CARs. The previous figure has been moved to the Supplementary Figure 6 for reference.
Q5;
The authors should include a discussion of their results in the context of other studies that did examine position effects when combining activation and co-stimulatory sequences, for example Maher et al Nat Biotech 2002; Finney et al JI 1998.
A5;
We have added in the text other studies that have examined the positional effects of combining activation and co-stimulation sequences.
Page 15, lines 432-441;
In earlier CAR studies, it has been reported that the order of CD3z-STD and CD28-STD affects the effector function of CAR-T cells. Although it is not possible to determine from these studies whether the functional strength of CAR-T cells was affected by the efficiency of CAR expression or STD structure, these studies are important to support our findings.
Q6;
Line 82: The abbreviation V/28/28/3z is first used here, but it isn’t clear here what the “V” stands for. The antigen specificity (VEGFR) should be described here for clarity.
A6;
We apologize for the lack of sufficient description. “V” in CAR ID stands for anti-VEGFR2 scFv. The text has been revised as follows. This scFv was generated from an antibody that specifically recognizes VEGFR2 (hybridoma; clone Avas12). We have not analyzed the antigen recognition properties of this scFv in detail, but we expect that it does not bind to other VEGFRs as well as the original antibody.
Page 3, lines 81-82;
We have previously shown that the vascular endothelial growth factor receptor 2 (VEGFR2)-specific first-generation CAR (V/28/28/3z) with CD28-HD/TMD is stably ex-pressed as a homogeneous molecule on mouse T cell membranes [11].
Q7;
Line 212: Exhaustion was not measured in this study, and this comment is a bit speculative. Cytotoxicity assay is only 18 hours, and not long enough to demonstrate exhaustion.
A7;
Thank you for pointing this out. The text has been revised to the following.
Page 7, lines 216-218;
(Before revision)
These results suggest that second-generation CARs with STDs derived from members of the TNFRSF have a high activation threshold compared with first-generation CARs, but can exert strong cytotoxic activity under conditions of a high E/T ratio by preventing T cell exhaustion and secreting large amounts of cytokines.
(After revision)
These results suggested that the CAR ability to induce CD3z-STD signaling-mediated cytotoxic activity was altered by differences in the type of 2nd STD (structure and signal input mechanism).
Q8;
Line 252-254: First sentence states “none of the second gen CAR-T cells showed antigen-independent proliferation”, however V/28/28/278-3z and V/28/28/137-3z do show significant proliferation in Figure 3C. Please clarify.
A8;
Thank you for pointing this out. The text has been revised to the following.
Page 7, lines 216-218;
(Before revision)
We evaluated the proliferative activity of second-generation CAR-T cells lacking the 2nd STD signal, and found that none of the second-generation CAR-T cells showed antigen-independent proliferation (Figure 3C).
(After revision)
We evaluated the proliferative activity of second-generation CAR-T cells lacking the 2nd STD signal, and found that none of the T cells expressing deficient in the 2nd STD signal input motif showed antigen-independent proliferation (Figure 3C).
Q9;
Line 353-354: Position of 2nd STD only effects cytotoxicity and IFNg secretion of CD28 family STD CARs, but not TNFRSF STD CARs. This conclusion cannot be generalised to all co-stimulatory sequences. Same again in the discussion, lines 473-475.
A9;
Thank you for pointing this out. The text has been revised to the following.
Page 12, lines 363-366;
(Before revision)
Taken together, these results indicated that CAR-mediated tonic signaling and enhanced secretion of TNF-a and IL-2 in CAR-T cells were based on the signal input of the 2nd STDs. Moreover, the position of the 2nd STD in the second-generation CAR had a significant effect on the IFN-g secretion and cytotoxic activity of CAR-T cells, suggesting that addition of the 2nd STD after the TMD may change the signal input efficiency of CD3z-STD depending on its structure.
(After revision)
Taken together, these results indicated that the enhanced secretion of IFN-g in CAR-T cells was based on the lengthening of the whole STD associated with the insertion of the 2nd STD, while the CAR-mediated tonic signaling and the enhanced secretion of TNF-a and IL-2 were based on the signaling input of the 2nd STD. Furthermore, the position of the 2nd STD in second-generation CARs had a significant effect on the cytotoxic activity of CAR-T cells, suggesting that addition of the 2nd STD after TMD may alter the signal input efficiency of the CD3z-STD depending on its structure.
Q10;
Line 427-428: I do not agree with this statement. There is wide appreciation that signalling from the native TCR is much more nuanced and regulated than signalling from CARs and there is little evidence that force or conformation play a role in translating antigen binding into ITAM phosphorylation.
A10;
Strictly speaking, the signal input mechanisms of CAR and TCR are different, but they both share the fact that the exposure of ITAMs in CD3 to the cytoplasm triggers the signal input for T cell activation. Then, studies on the TCR suggest that the cytoplasm of ITAMs in CD3 is released from the membrane by horizontal mechanical stimulation of the receptor. It is assumed that CAR, like TCR, promotes the exposure of CD3z-STD to the cytoplasm by binding the antigen recognition domain and dynamically changing the structure of the receptor. Our findings suggested that the efficiency of this CD3z-STD exposure is affected by the structure of the STD.
Q11;
Line 463-465: These reagents are widely available. If the analysis in question is beyond the scope of this study the authors should simply highlight a biochemical analysis of signalling pathways as a useful further analysis.
A11;
Thank you for pointing this out. The text has been revised to the following.
Page 17, lines 499-500;
(Before revision)
In this study, we could not determine the effect on CAR signaling of adding the 2nd STD because we could not get the appropriate antibodies to analyze the activation signals in mouse T cells.
(After revision)
We believe that further analysis of CARs with artificial 2nd STD without signal input motifs, as well as biochemical analysis of signaling pathways and structural analysis of intracellular regions in second-generation CARs, will help us to elucidate the elements of STD that affect the signal input efficiency of CARs as shown in Figure 5B.
Q12;
Line 470-471: This statement only applies to the CD137-derived STD and should not be generalised.
A12;
Thank you for pointing this out. The text has been revised to the following.
Page 16, lines 499-500;
(Before revision)
Moreover, while the co-stimulatory signals of the 2nd STD enhanced the secretion of TNF-a and IL-2 in T cells depending on their type, ...
(After revision)
Moreover, while the co-stimulatory signals of the CD28-STD, CD278-STD, and CD137-STD enhanced the secretion of TNF-a and/or IL-2 in T cells depending on their type, ...
Again, thank you for giving us the opportunity to strengthen our manuscript with your valuable comments and queries. We have worked hard to incorporate your feedback and hope that these revisions persuade you to accept our submission.
Round 2
Reviewer 2 Report
The authors have carefully addressed all concerns raised and significantly adapted the manuscript and data presentation. Currently, this work is quite clear and may help other researchers in the field shape their own research based on the findings herein presented. Congratulation on the fast and precise work.
Reviewer 3 Report
The authors have provided adequate responses and appropriate edits, with inclusion of the additional data presentations and references where requested.
A minor final addition: the authors should note in the figure legends where non-normalised data graphs are provided in supplement.